# Association analyses of host genetics, root-colonizing microbes, and plant phenotypes under different nitrogen conditions in maize

Michael A Meier[1,2], Gen Xu[1,2], Martha G Lopez-Guerrero[3], Guangyong Li[1,2], Christine Smith[2], Brandi Sigmon[4], Joshua R Herr[2,4], James R Alfano[2,4†], Yufeng Ge[5], James C Schnable[1,2]*, Jinliang Yang[1,2]*

[1]Department of Agronomy and Horticulture, University of Nebraska-Lincoln, Lincoln, United States; [2]Center for Plant Science Innovation, University of Nebraska-Lincoln, Lincoln, United States; [3]Department of Biochemistry, University of Nebraska-Lincoln, Lincoln, United States; [4]Department of Plant Pathology, University of Nebraska-Lincoln, Lincoln, United States; [5]Biological Systems Engineering Department, University of Nebraska-Lincoln, Lincoln, United States

**\*For correspondence:**
schnable@unl.edu (JCS);
jinliang.yang@unl.edu (JY)

[†]Deceased

**Competing interest:** The authors declare that no competing interests exist.

**Abstract** The root-associated microbiome (rhizobiome) affects plant health, stress tolerance, and nutrient use efficiency. However, it remains unclear to what extent the composition of the rhizobiome is governed by intraspecific variation in host plant genetics in the field and the degree to which host plant selection can reshape the composition of the rhizobiome. Here, we quantify the rhizosphere microbial communities associated with a replicated diversity panel of 230 maize (*Zea mays* L.) genotypes grown in agronomically relevant conditions under high N (+N) and low N (-N) treatments. We analyze the maize rhizobiome in terms of 150 abundant and consistently reproducible microbial groups and we show that the abundance of many root-associated microbes is explainable by natural genetic variation in the host plant, with a greater proportion of microbial variance attributable to plant genetic variation in -N conditions. Population genetic approaches identify signatures of purifying selection in the maize genome associated with the abundance of several groups of microbes in the maize rhizobiome. Genome-wide association study was conducted using the abundance of microbial groups as rhizobiome traits, and n=622 plant loci were identified that are linked to the abundance of n=104 microbial groups in the maize rhizosphere. In 62/104 cases, which is more than expected by chance, the abundance of these same microbial groups was correlated with variation in plant vigor indicators derived from high throughput phenotyping of the same field experiment. We provide comprehensive datasets about the three-way interaction of host genetics, microbe abundance, and plant performance under two N treatments to facilitate targeted experiments toward harnessing the full potential of root-associated microbial symbionts in maize production.

## Editor's evaluation

It is widely assumed that plants actively manipulate their root associated microbiomes although the genetic factors that contribute to this process have yet to be discovered. The authors catalog the root-associated bacteria that associate with diverse maize lines, grown under low and high nitrogen treatments and through a genome-wide association study, identify candidate genetic loci that influence the plant associated microbiome. In addition to the interesting insights reported in

the present manuscript, these data are likely to be used for and/or compared to, in many future studies.

## Introduction

Symbiotic relationships between plant hosts and root-associated microbes have been shaped through natural selection over millions of years of coevolution (*Limborg and Heeb, 2018*), and have been a driver of crop performance and yield in agricultural production since the beginning of plant domestication (*Yadav et al., 2018*). Microbial actors in the rhizosphere have been shown to promote plant growth (*Saleem et al., 2019*), improve nutrient use efficiency (*Gomes et al., 2018*; *Zhu et al., 2016*), and reduce abiotic stress response (*Hussain et al., 2018*). The promise of high throughput screens for plant growth promoting activity in isolated microbial strains or synthetic communities (*Singer et al., 2021*; *Yee et al., 2021*) is the potential discovery of microbial agents that can be used as seed or soil additives to improve crop performance under field conditions. Promising results have been observed in controlled environments (*Van Gerrewey et al., 2020*; *Xi et al., 2020*; *Yu et al., 2021*), but it remains a challenge to achieve similar outcomes in crops under agriculturally relevant field conditions (*Eida et al., 2017*; *Kaur et al., 2020*; *Sessitsch et al., 2019*), as many microbial inoculants struggle to compete with naturally occurring microbes in the rhizosphere and rarely survive for extended periods of time in the field (*Piromyou et al., 2011*). An improved understanding of how plants shape the composition of their rhizobiomes under diverse field conditions would make it more feasible to identify beneficial plant-microbe interactions that will be persistent and replicable in field environments. Moreover, studying the effects of plant genetics on microbial communities may identify opportunities to breed crop plants that recruit and maintain superior growth-conducive microbial communities from the natural environment.

Few studies to date have addressed the relationship between plant genetics and rhizobiomes in field settings, mainly because large-scale rhizosphere sampling (as opposed to leaf microbiome sampling) and DNA sequence analysis of microbial communities in diverse plant hosts is time-consuming, expensive, and poses significant logistical and technical challenges. It has been shown that plant genetics can explain variation in both root architecture (*Bray and Topp, 2018*) and exudation (*Mönchgesang et al., 2016*). If these factors in turn shape microbial communities (*Sasse et al., 2018*), variation in the root-associated microbial groups (hereafter referred to as rhizobiome traits) may also result from genetic factors. Recent studies suggested that the variation in the composition of rhizobiomes is likely controlled by plant genetic factors (i.e., heritable) in maize (*Peiffer et al., 2013*), sorghum (*Deng et al., 2021*), and switchgrass (*Sutherland et al., 2021*). However, it remains unclear to what extent these heritable microbes are affected by the plant host and contribute to variation in the crop phenotype. Like any other trait under heritable genetic control, rhizobiome traits can be targeted in selective breeding experiments. To explore this idea, previous efforts have been directed towards identifying plant genetic loci that are associated with rhizobiome traits. Initial studies have shown that rhizosphere microbial communities differ between distinct genotypes of the same host species, which has been shown in a study on 27 maize genotypes (*Peiffer et al., 2013*; *Walters et al., 2018*) and more recently, in a diversity panel of 200 sorghum lines (*Deng et al., 2021*). Genome-wide association study (GWAS) has successfully revealed associations between plant genes and rhizobiome traits at high-level measures of rhizosphere community dissimilarity (i.e. using principal components) in an *Arabidopsis* diversity panel (*Bergelson et al., 2019*) or at order level (derived from operational taxonomic units [OTUs]) in a sorghum diversity panel (*Deng et al., 2021*). However, previous attempts at linking plant genes to the abundance of specific groups of microbes have had limited success due to small population size, limited host genetic diversity, or due to insufficient taxonomic resolution (*Beilsmith et al., 2019*; *Liu et al., 2021*). It was observed previously (*Zhu et al., 2016*) that soil microbial communities drastically change in response to N fertilization. In bulk soil, this is likely due to a direct effect of N application or lack thereof. In rhizospheres, however, only a subset of the observed changes can be attributed to direct effects of nitrogen (N) fertilization, while particular microbial groups may be subject to indirect effects induced by the plant host in response to N availability or deficiency (*Meier et al., 2021*). A possible explanation for this could be that during most of the interval between maize domestication and the present, beneficial plant-microbe interactions have evolved in low-input agricultural systems characterized by relative scarcity of nutrients, predominantly

nitrogen (*Brisson et al., 2019*). This is in stark contrast to the modern agricultural environment that has been the norm since the 1960s, in which plants are supplied with large quantities of inorganic N fertilizer (*Cao et al., 2018*). As a consequence, previous selection pressure to retain traits favorable under low N conditions, including plant growth-promoting microbes, has been largely reduced in modern maize breeding programs (*Haegele et al., 2013*; *Zhu et al., 2016*). Thus, if a microbial group is indeed under host genetic control and has an effect on plant fitness (i.e. promotes plant development or increases crop yield) under either N condition, we would expect the rhizobiome trait to be under host selection.

In the present study, we evaluate the role that selection on plant genetic factors has played in shaping the maize rhizobiome under different N conditions. We employ the Buckler-Goodman maize diversity panel, a set of maize lines selected for maximum representation of genetic diversity and growth in temperate latitudes (*Flint-Garcia et al., 2005*). This population has previously been used to determine the heritability of leaf microbiome traits and to perform genome-wide association studies (GWAS) on a number of different phenotypes (*Wallace et al., 2018*). We collected replicated data on the rhizobiome of 230 lines drawn from this panel when grown under either high nitrogen (+N) and low nitrogen (-N) conditions in the field. For 150 microbial groups present in the rhizosphere (referred to as 'rhizobiome traits'), which were abundant and consistently reproducible, we quantify the degree to which variation is subject to plant genetic control, and test for evidence of selection under either or both N conditions. Using a set of 20 million high-density single-nucleotide polymorphisms (SNPs), we perform GWAS for each rhizobiome trait identifying genomic loci that are associated with one or more rhizobiome traits. Through comparison with gene expression data generated for the same population, we determine whether genes near microbe-associated plant loci are preferentially expressed in root tissue. Lastly, we evaluate whether the abundance of each microbial group in the rhizosphere is correlated with plant performance traits measured in the field, and whether microbe abundance and plant performance depend on the allele variant at selected microbe-associated plant loci. The results presented in this study lay the groundwork for future endeavors to investigate the molecular mechanisms of specific plant-microbe interactions under agronomically relevant conditions.

## Results

### Characterization of the rhizobiome for diverse maize genotypes under two different N conditions

3,313 rhizosphere samples from 230 replicated genotypes of the maize diversity panel (*Flint-Garcia et al., 2005*) were collected from field experiments conducted under both +N and -N conditions (Materials and methods). At the time of sampling, visible phenotypic differences were observable between +N and -N plots as measured through aerial imaging (details are reported in *Rodene et al., 2022* using the same experimental field). Paired-end 16 S sequencing produced 216,681,749 raw sequence reads representing 496,738 unique amplicon sequence variants (ASVs) (Materials and methods). Raw reads were subjected to a series of quality checks and abundance filters following a workflow for 16 S sequencing data analysis by *Callahan et al., 2016a*, which resulted in a curated dataset of 3626 ASVs for 3306 samples, and 105,745,986 total ASV counts (*Supplementary file 1*). This dataset includes ASVs that are highly abundant across the maize diversity panel and reproducible in both years of sampling. Constrained Principal Coordinates analysis calculated from the abundances of 3626 ASVs shows divergence of samples collected under either -N or +N treatment (*Figure 1A*), which indicates that the microbiomes differ between these two experimental conditions (PERMANOVA p-value for N treatment <0.001).

An initial analysis looking at high-level rhizobiome traits (Principal Components and alpha diversity metrics derived from the ASV table) shows the same pattern of divergent microbial communities between N treatments, and in particular under the -N treatment there is evidence for the association of plant genomic loci and microbiome composition (*Figure 1—figure supplement 1*). To study changes in rhizobiome composition more accurately, the final 3626 ASVs were clustered into n=150 distinct microbial groups ('rhizobiome traits'), spanning 19 major classes of rhizosphere microbiota (*Figure 1B*, *Supplementary files 2 and 3*) using a method previously described (*Meier et al., 2021*, **Supplementary Methods**). Of these rhizobiome traits, 79/150 (52.7%) groups were significantly more abundant in samples collected from the +N condition (t-test, p<0.05), 53/150 (35.3%) significantly

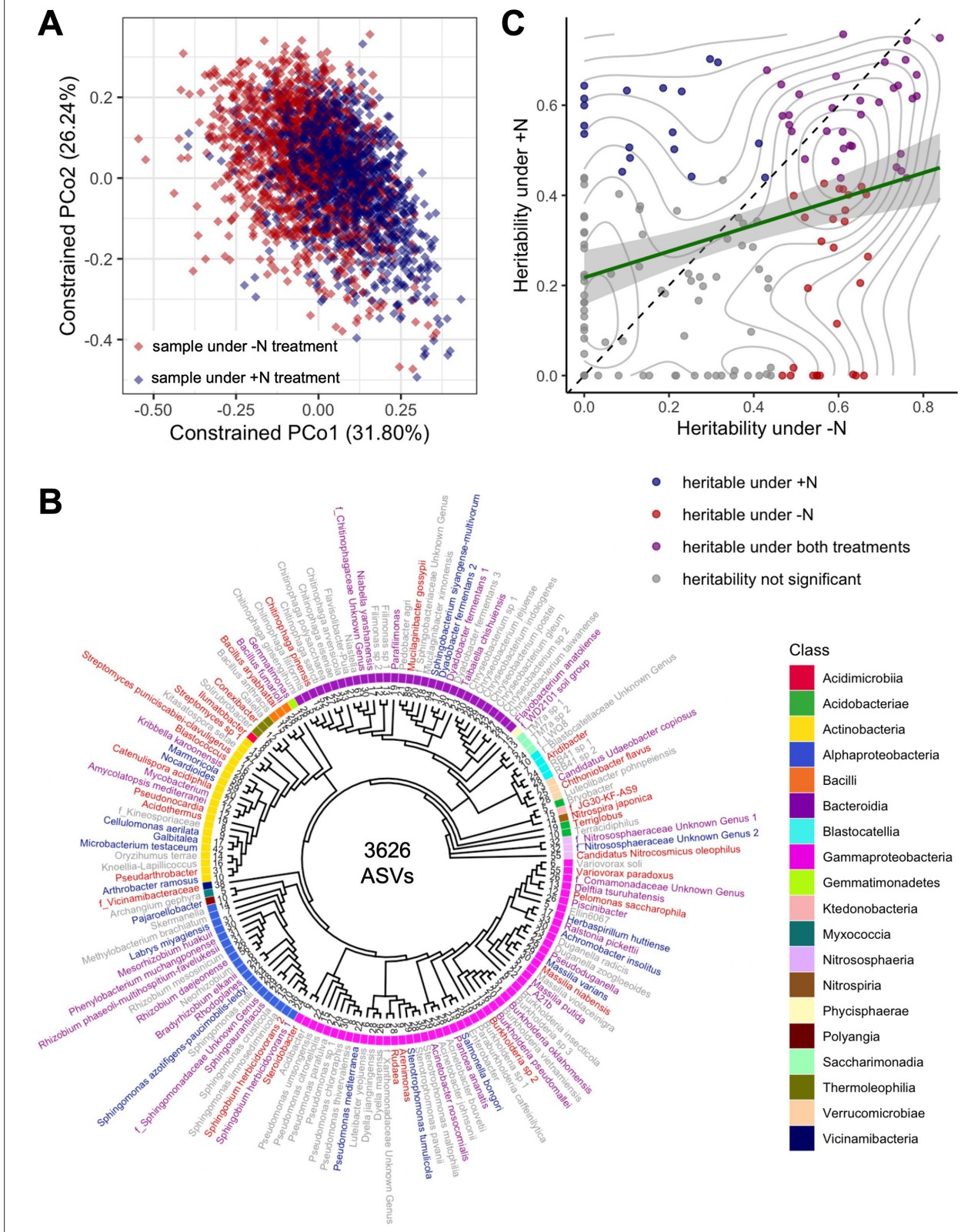

**Figure 1.** Diversity, phylogenetics, and heritability of rhizobiome traits. (**A**) Constrained ordination analysis showing the largest two principal coordinates calculated from the abundances of 3626 ASVs. Each diamond represents one sample collected from plants under +N (blue) and -N (red) treatment, respectively. Note the separation of N treatments along PCo1. (**B**) Phylogenetic tree of 150 taxonomic groups of rhizosphere microbiota ('rhizobiome traits') generated by clustering 3626 ASVs. Families are prefixed with 'f_', genus and species names are given where known. Numbers at tree tips

*Figure 1 continued*

indicate distinct ASVs in each group. Label colors indicate heritability of each rhizobiome trait as in panel C. (**C**) Heritability (**h**$^2$) calculated for all 150 rhizobiome traits under +N and -N treatments. Green line indicates linear regression with 95% confidence interval, $r^2$=0.104. Diagonal dashed line denotes identity. Gray lines mark density of data points. Colors indicate whether traits are significantly heritable under either or both N treatments, as determined through a permutation analysis using 1000 permutations.

The online version of this article includes the following figure supplement(s) for figure 1:

**Figure supplement 1.** GWAS of high-level rhizobiome traits.

**Figure supplement 2.** Abundance and heritability of 150 microbial groups.

**Figure supplement 3.** Annotations of heritable microbial groups.

more abundant in samples collected from the -N condition, and 18/150 (12.0%) showed no significant difference in abundance between the two treatments. In several cases, more closely related microbial groups exhibit shared patterns of differential abundance between N treatments (*Figure 1—figure supplement 2A*).

## Rhizobiome traits are more heritable under -N conditions

The abundance of each of the 150 rhizobiome traits was assessed separately for +N and -N conditions, and the heritability (proportion of total variance explicable by genetic factors) was estimated using an approach following a previous study (*Deng et al., 2021*) (Materials and methods). Rhizobiome traits were comparatively more heritable under -N than +N conditions (paired Student's t-test, p=0.021, *Figure 1C*). We found 34/150 (22.7%) microbial groups to be significantly heritable (permutation test, p<0.05, Materials and methods) under both N conditions, 18/150 (12%) only under +N conditions, and 27/150 (18%) only under -N conditions. Twelve rhizobiome traits exhibited estimated h$^2$ >0.6 in both +N and -N conditions (*Figure 1—figure supplement 3*). These include four groups of ASVs assigned to the order *Burkholderiales* (the genus *Pseudoduganella*, an unknown genus in the *Comamonadaceae* family, the family *A21b*, and *Burkholderia oklahomensis*) and two groups in the *Sphingomonadales* order (*Sphingobium herbicidovorans 1* and an unknown genus in the *Sphingomonadaceae* family). Notably, closely related microbial groups did not exhibit obvious patterns of shared high or low estimated heritabilities (*Figure 1B*). As heritabilities and responses to treatments can vary considerably within families, genera, and lower taxonomic ranks, this underscores the importance of adequate taxonomic resolution when analyzing rhizosphere microbial communities. We further observed that more abundant microbes in the rhizosphere also tend to be more heritable. The correlation of relative abundance vs. heritability was $r$=0.29 (Pearson's correlation test, p=3.4 × 10$^{-4}$) for +N and $r$=0.39 (Pearson's correlation test, p=1.1 × 10$^{-6}$) for -N (*Figure 1—figure supplement 2B*).

## Rhizobiome traits are related with plant fitness and predominantly under purifying selection

Under the hypothesis that the rhizobiome traits have effects on plant fitness, we sought to estimate the selection differentials under different N treatments (*Rausher, 1992*). To reduce biases due to environmental covariances (*Lande and Arnold, 1983*), the standardized BLUP values of the microbial traits were fitted into the fitness function (See Materials and methods). For the selection differential estimation, the canopy coverage (CC) obtained from UAV imaging was used as a proxy for plant fitness. As a result, we identified 58 unique rhizobiome traits exhibiting significant linear selection differentials (bootstrapping p-value <0.05) under +N (28 traits) and -N (46 traits) treatments (*Figure 2—figure supplement 1*). Additionally, four rhizobiome traits showed significant quadratic selection differentials (+N: *Luteolibacter pohnpeiensis* [–2.627913e-05, p-value = 0.044], -N: *Blastococcus* [8.516159e-06, p-value = 0.03], *Pseusomonas umsongensis* [–2.003792e-05, p-value = 0.04], *Chthoniobacter flavus* [–5.807404e-05, p-value = 0.028]).

Selection acting on rhizobiome traits can happen either by purging deleterious alleles (purifying selection) or by elevating the frequencies of advantageous alleles (positive selection). To evaluate the mode of selection at the genomic level, a Bayesian-based method (Genome-wide Complex Trait Bayesian analysis, or GCTB) was used to test for each rhizobiome trait (Materials and methods). A set of n=834,975 independent SNPs was used to estimate their effects on 150 rhizobiome traits as well as 17 conventional plant traits collected from the same population in the same field experiments

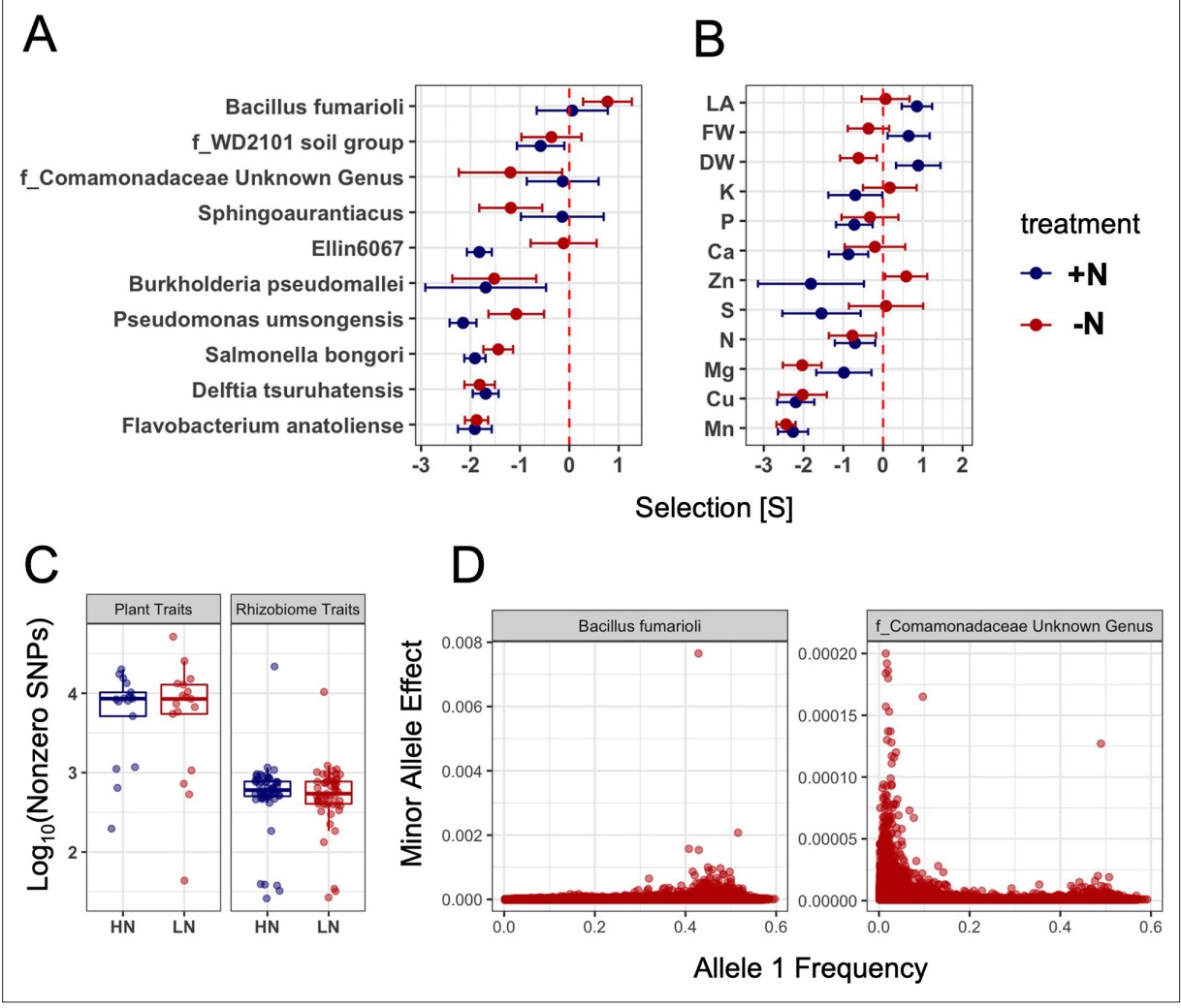

**Figure 2.** Population parameters estimated from genome-wide SNPs for plant and rhizobiome traits. Selection coefficients (S value) of rhizobiome (**A**) and plant (**B**) traits calculated for both N treatments using genome-wide independent SNPs. A negative S value indicates negative (purifying) selection, and a positive S value indicates positive (directional) selection. Traits are shown that show significant selection under one or both N treatments. (**C**) Number of SNPs showing non-zero effects for both plant and rhizobiome traits. (**D**) Examples of positive (*Bacillus fumarioli*) and purifying selection (*f_Comamonadaceae Unknown Genus*) showing minor allele effect vs. allele 1 frequency with data skew to the right and to the left, respectively.

The online version of this article includes the following figure supplement(s) for figure 2:

**Figure supplement 1.** Rhizobiome traits exhibit significant linear selection differentials (bootstrapping p-value <0.05) under +N and -N treatments.

(Materials and methods, ***Supplementary file 4***). Using the relationship between effects of non-zero SNPs and their minor allele frequencies (MAFs) as a proxy for the signature of selection (***Zeng et al., 2018***), the S parameter was jointly estimated from the GCTB analysis for rhizobiome traits and plant traits. According to Zeng (***Zeng et al., 2018***), if S=0 (i.e. the posterior distribution of S is insignificantly different from zero), the SNP effect is independent of MAF, suggesting neutral selection. If there is selection acting on the trait, the SNP effect can be positively (S>0) or negatively (S<0) related to MAF, indicating positive and purifying selection, respectively.

We report 10 rhizobiome traits that showed both significant linear selection differentials and significant S parameters (***Figure 2A***). Under these stringent criteria, nine rhizobiome traits show evidence of purifying selection under +N or under -N. One microbial group (*Bacillus fumarioli*) showed positive S values indicating that this trait might have been a target of positive selection. Relative to rhizobiome traits, plant leaf traits and nutrient traits were both more likely to exhibit evidence of selection within this maize population. Three out of 15 plant leaf traits, that is leaf area (LA), leaf fresh weight (FW), and

leaf dry weight (DW) (Materials and methods), exhibited S>0 values under the +N condition, consistent with positive selection, while only one of the three exhibited a slightly negative S value in the -N condition and in that case exhibited a pattern consistent with weak purifying selection (*Figure 2B*). Note that the three leaf-related traits are not independent. The pairwise correlation coefficients are 0.92, 0.91, and 0.94, for LA and FW, LA and DW, FW and DW, respectively. Of the 11 micronutrient traits evaluated, 9/11 and 4/11 showed significantly negative S values in trait data collected under +N and -N conditions, respectively. From the same GCTB analysis, estimates of the number of SNPs with non-zero effects were substantially lower for rhizobiome traits than for conventional plant traits, whereas the differences were insignificant between the two N treatments for both rhizobiome and plant traits (*Figure 2C*). Using these non-zero effect SNPs, we plotted their minor allele frequency vs. the minor allele effect. As expected, in the case of positive selection (*Bacillus fumarioli*), we observed a skew towards higher MAF and in the case of purifying selection (*f_Comamonadaceae Unknown Genus*), a skew towards lower MAF (*Figure 2D*).

## Genes underlying microbe-associated plant loci are preferentially expressed in root tissue

The observation that many rhizobiome traits are both under significant host genetic control and targets of selection suggests it may be possible to detect individual large effect loci controlling rhizobiome traits. To investigate this, we performed GWAS using each of the 150 rhizobiome traits. This analysis was done separately for the -N and +N conditions, as N deficiency induces dramatic changes in plant metabolism, including changes in root gene expression (*Singh et al., 2013*) and root exudation (*Zhu et al., 2016*), and because N applied to the field directly impacts soil and rhizosphere microbiomes (*Meier et al., 2021*). We focused on 'hotspots' along the genome where we find the highest cumulative density of significant associations between SNPs and any rhizobiome traits under either N treatment, because morphological (i.e. root architecture) or physiological (root exudation) changes may simultaneously affect several rhizobiome traits. For this purpose, we split the maize genome into 10 kb genomic windows and tallied the number of significant ($p<10^{-7.2}$) GWAS signals in each window. This analysis revealed 622 genomic regions containing at least one significant GWAS signal, and we refer to these regions as microbe-associated plant loci (MAPLs) (Materials and methods). We report these MAPLs alongside nearby genes in *Supplementary file 5*. Out of 150 microbial groups, 104 were associated with at least one of the 622 loci.

To reduce false discoveries, we decided to discuss a subset of 119 MAPLs here, that had at least two significant GWAS signals. Among these 119 MAPLs, 69 were observed under +N treatment and 50 under -N treatment (*Figure 3A*, *Supplementary file 5*). Of the 150 rhizobiome traits evaluated here, 35 were associated with at least one of the 119 MAPLs, with 21 rhizobiome traits associated with 69 MAPLs under the +N treatment and 17 rhizobiome traits with 50 MAPLs under the -N treatment. Three rhizobiome traits (*f_Chitinophagaceae Unknown Genus*, *Sphingoaurantiacus*, and *f_Vicinamibacteraceae*) showed significant associations under both N treatments, albeit with different plant loci. No loci were found that had associations with rhizobiome traits under both N treatments, which is expected as GWAS analyses were done separately for different N treatments and the microbial groups studied here were partly distinguished based on differential abundance in response to N treatments.

We hypothesized that many plant genes underlying MAPL hotspots may exert control over the rhizosphere microbiome via changes in root physiology, architecture, and exudate composition (*Vandenkoornhuyse et al., 2015*) and may therefore be preferentially expressed in root tissue. Transcribed sequences of 97 gene models were completely contained within ±10 kb of the 119 MAPL hotspots identified here, where 114/119 MAPLs contained between 1 and 5 genes. We evaluated the expression of these MAPL genes relative to the overall patterns exhibited by all genes outside the MAPL regions in seven tissues using published expression data from the same maize population (*Kremling et al., 2018*). Expression data was available in this dataset for 73 out of 97 MAPL genes across 298 maize genotypes from tissue samples collected at germination and during flowering time. These MAPL genes, when compared to (n=29,771) other genes available in the dataset, show on average significantly higher expression in the germinating root, the germinating shoot, and the third leaf base (*Figure 3B*).

To complement the gene expression data provided by Kremling et. al, we selected four diverse and well characterized maize genotypes (K55, W153R, B73, and SD40). Plants were grown in a controlled

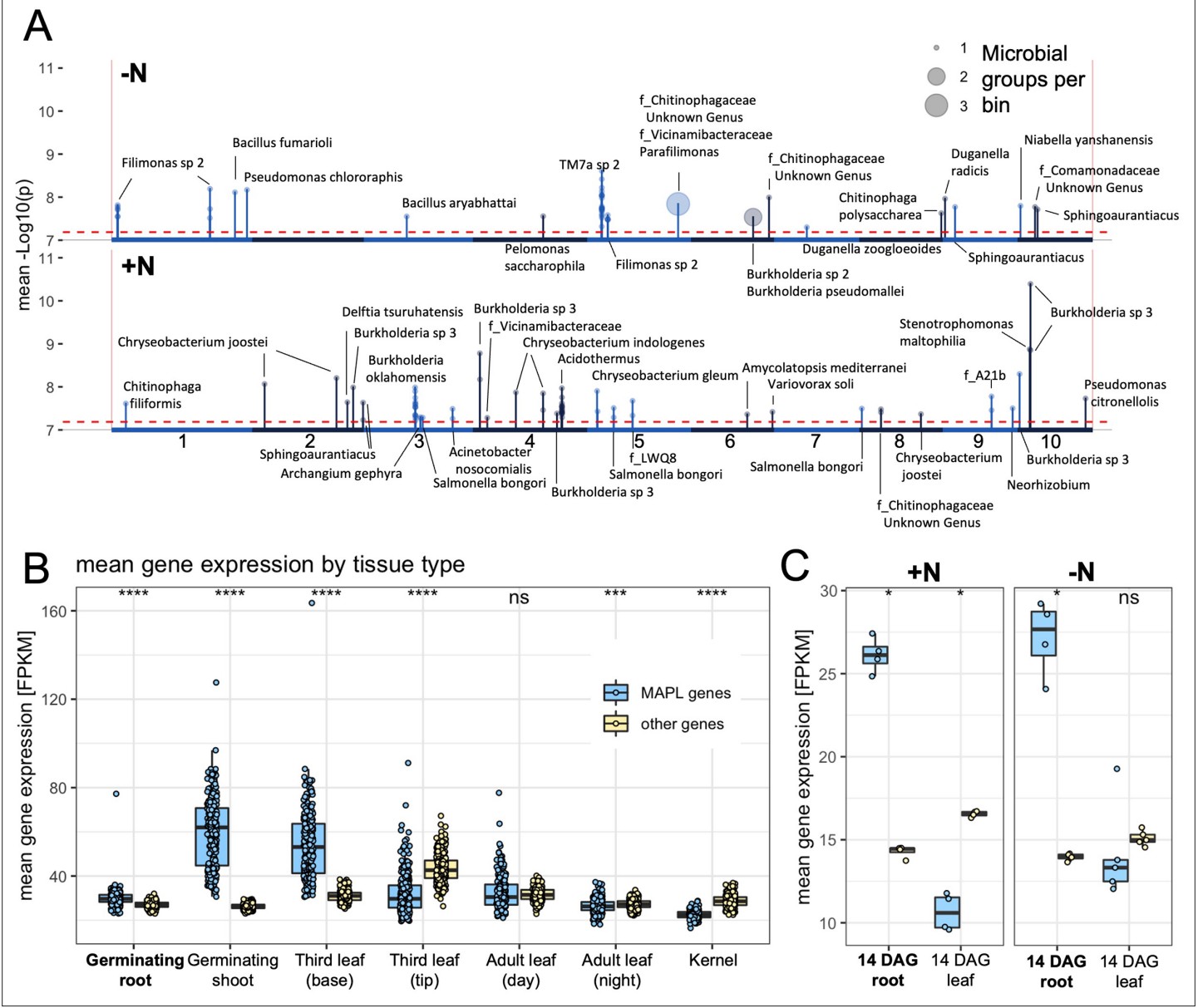

**Figure 3.** Microbe associated plant loci (MAPLs) contain genes expressed in roots. (**A**) GWAS of 150 rhizobiome traits reveals microbe-associated plant loci across the maize genome. Dashed line indicates the -log10(p)=7.2 significance level for GWAS signals. Circles on top of peaks at each MAPL indicate the number of rhizobiome traits associated with each locus. Each MAPL is annotated with the associated rhizobiome trait(s) that showed significant GWAS signals. (**B**) Mean gene expression of 73 MAPL genes and 29,771 other genes in seven tissue types, measured in 298 genotypes of the maize diversity panel (***Kremling et al., 2018***). (**C**) Mean gene expression of 97 MAPL genes and 44,049 other genes in two tissue types, measured in the present study in four maize genotypes under +N and -N treatments.

greenhouse environment under standard N and N deficient conditions and gene expression was analyzed in roots and shoots of two-week old seedlings (for details refer to ***Xu et al., 2022***). In agreement with the dataset provided by Kremling et al, significantly higher expression of 97 MAPL genes was observed in root but not leaf tissue compared to (n=44,049) other genes available in this dataset (***Figure 3C***). No strong physiological response to N deficiency was expected for 2-week-old seedlings and no significant differences were observed in the pattern of MAPL gene expression between the two N treatments.

Collectively, these data are consistent with the hypothesis that rhizobiomes are at least in part genetically controlled by the host plant in a process mediated by plant gene expression.

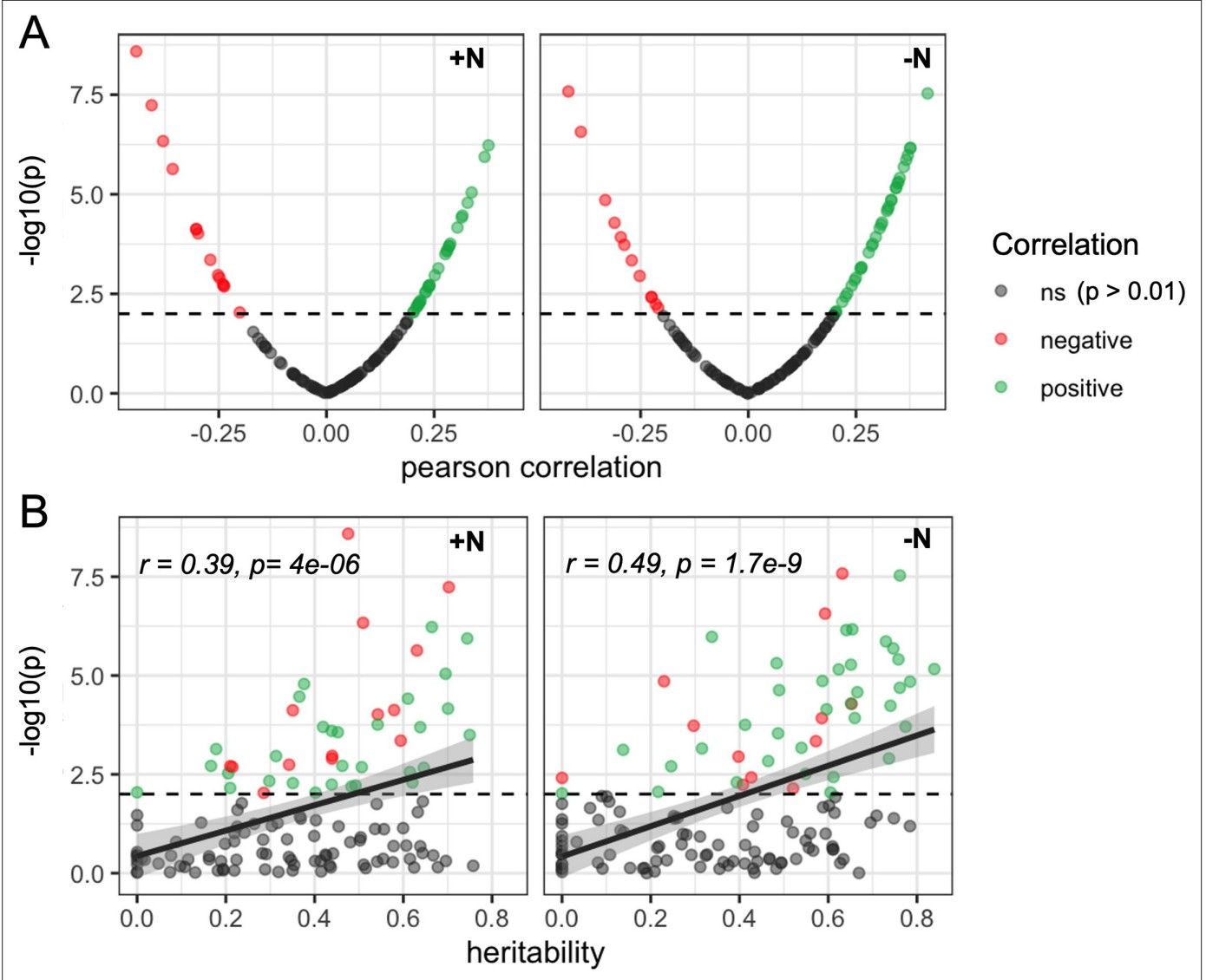

**Figure 4.** Heritable microbial groups tend to be correlated with whole plant canopy coverage. (**A**) Distribution of statistical significance and correlation values for the relationship between canopy coverage (CC) and each of 150 microbial groups under either +N or -N conditions. Dashed line indicates significance level (p=0.01). (**B**) Relationship between the estimated heritability of individual rhizobiome traits and correlation of the same individual rhizobiome traits with variation in CC. Dashed line indicates significance level (p=0.01).

The online version of this article includes the following figure supplement(s) for figure 4:

**Figure supplement 1.** Correlation of microbe abundance with 17 agronomic and micronutrient traits under +N (blue) and -N (red) conditions.

**Figure supplement 2.** Microbial traits that correlate with canopy coverage.

## Heritable and adaptively selected rhizobiota are associated with plant phenotypes

We investigated the correlation of microbe abundance with 17 plant traits, including leaf physiology, leaf micronutrient traits, and traits extracted from aerial images (Materials and methods) to identify potential plant phenotypic consequences of variation in the abundance of specific rhizosphere microbes. Several rhizobiome traits were significantly correlated (p<0.01) with measures of plant performance, such as leaf area, leaf dry weight and fresh weight, and with several measures of leaf micronutrients such as nitrogen, sulfur, and phosphorus (*Figure 4—figure supplement 1*). The trait that was most strongly linked to microbe abundance was leaf canopy coverage (CC). In total, 62 microbial groups – more than expected by chance (permutation test, p<0.001) – were significantly

(Pearson correlation test, p<0.01) associated with CC (marked in *Figure 4* in green for positive correlation and in red for negative correlation). 30 microbial groups under +N and 35 under -N were positively correlated with CC. 14 groups under +N and 12 under -N were negatively correlated with CC. 15 microbial groups were associated with CC under +N treatment, 18 under -N treatment, and 29 showed a significant association under both N treatments (*Figure 4A*). Under both N treatments, we observe an association between heritability and the correlation with CC, which was statistically significant (Pearson correlation coefficient $r$=0.39, p=4 × 10$^{-6}$) for +N and even more significant ($r$=0.49, p=1.7 × 10$^{-9}$) under the -N condition (*Figure 4B*).

We summarize the relationship of the analyses conducted in this study under either N treatment for the 62 microbial groups that are correlated with CC. 44/62 (71%) are heritable and 13/62 (21%) are under selection under either or both N treatments (*Figure 4—figure supplement 2* ). 56/62 (90%) show strong GWAS signals in 174/467 (39%) of the MAPLs identified here, which contain 255/395 (65%) of possibly microbe-associated genes. Four microbial groups, *Sphingoaurantiacus*, *Bacillus fumarioli*, *f_Comamonadaceae* Unknown Genus, and *Burkholderia pseudomallei* are of particular interest as they overlap in all performed assays, showing evidence of heritability and selection, a strong GWAS signal in associated plant genomic loci, and positive correlation with canopy coverage. The complete summary data for all 150 microbial groups are available in *Supplementary file 3*.

Overall, our data show a clear trend that the 62 microbial groups associated with plant performance also tend to be associated with host genetics, and the datasets provided here can be used to design more targeted experiments to confirm associations of rhizosphere microbial groups with plant genetics and performance on a case-by-case basis.

## Allelic differences at microbe-associated plant loci predict microbe abundance

We identified several strong GWAS signals that link plant genomic loci to rhizobiome traits (*Figure 3A*). Such signals indicate that the pattern of SNPs at a given locus (i.e. the genetic architecture) has a large magnitude of effect attached to the abundance of the associated microbial groups. Next, we sought to determine whether a particular allele (either the major or the minor variant) in our maize population is associated with an increased or decreased abundance of the corresponding microbe.

The unknown genus in the *Comamonadaceae* family mentioned above, while unnamed and uncharacterized, shows high heritability under both N treatments (h$^2$=0.610 under +N, and 0.651 under -N, *Figure 1B and C*), and shows evidence of being under purifying selection under -N (*Figure 2A and D*). Under the same environmental conditions, a significant MAPL controlling variation in microbial abundance is detectable on maize chromosome 10 (*Figure 3A* and *Figure 5A*). This same rhizobiome trait is among those that are positively correlated with CC under both -N ($r$=0.347, p=5.313 × 10$^{-6}$) and +N ($r$=0.314, p=3.845 × 10$^{-5}$) (*Figure 4A*). A total of five annotated gene models are located near the peak of significant SNP markers that define the chromosome 10 MAPL for this rhizobiome trait (*Figure 5A and B*). A linkage disequilibrium block was observed between 23.90 and 23.96 MB on maize chromosome 10, spanning the set of significantly associated SNPs and three candidate genes Zm00001d023838, Zm00001d023839, and Zm00001d023840 (*Figure 5C*). In accordance with *Figure 3C*, these genes are preferentially expressed in roots (*Figure 5—figure supplement 1*). As described above, the abundance of the *f_Comamonadaceae* genus was significantly correlated with variation in CC, shown here for the -N treatment (*Figure 5D*). Next, we used the haplotype information at the target SNP to mark genotypes that carry the major allele or the minor allele, respectively, and the abundance of the *f_Comamonadaceae* genus was significantly higher in rhizosphere samples collected from maize genotypes carrying the major allele than in samples collected from maize genotypes carrying the minor allele (*Figure 5E*). However, CC was not significantly different between maize genotypes carrying either the major or minor allele of the chromosome 10 MAPL (*Figure 5F*).

The example discussed here shows a three-way association of the abundance of a particular microbial group in the rhizosphere, a corresponding locus on the maize genome, and plant performance in the field. The datasets provided alongside this publication contain several such associations and may serve as the basis for more targeted experiments to establish a direction of causation between microbe abundance and plant performance, and to shed light on the genetic mechanisms that shape symbiotic relationships between the plant host and associated rhizosphere microbes.

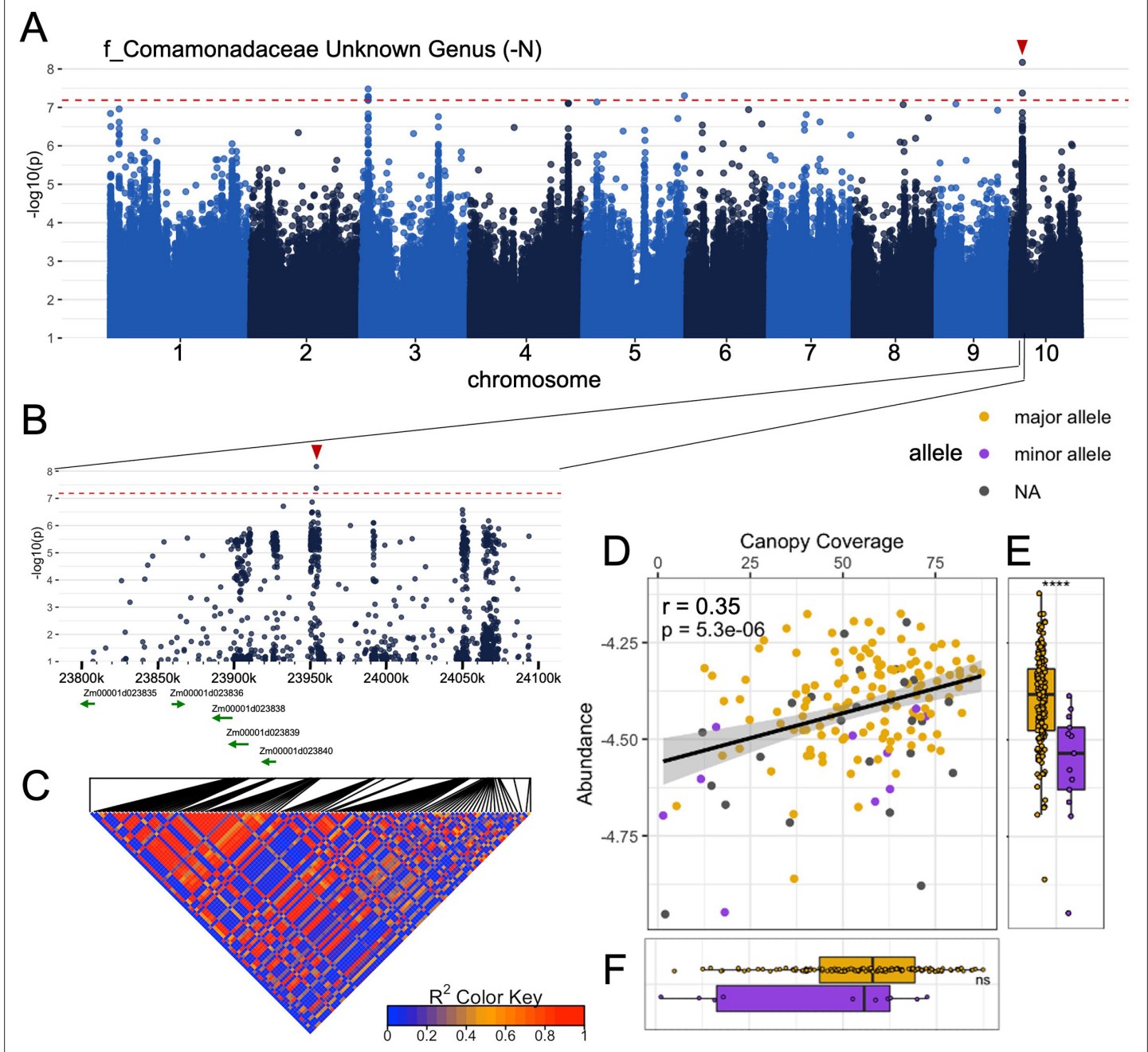

**Figure 5.** Abundance of heritable, adaptively selected microbes depends on allelic differences at MAPLs. (**A**) Results of a genome wide association study conducted using values for the rhizobiome trait (*f_Comamonadaceae Unknown Genus*) observed for ~230 maize lines grown under nitrogen deficient conditions. Alternating colors differentiate the 10 chromosomes of maize. Dashed line indicates a statistical significance cutoff of -log10(p)=7.2. (**B**) Zoomed in visualization of the region containing the peak observed on chromosome 10. (**C**) Linkage disequilibrium among SNP markers genotyped within this region, calculated using genotype data from 271 lines (**D**) Correlation plot of microbe abundance vs. canopy coverage (CC). Each point represents a maize genotype. Differences in microbe abundance (**E**) and CC (**F**) are marked between genotypes carrying the major allele (gold) vs the minor allele (purple) at the target SNP (red arrow in panel A and B).

The online version of this article includes the following figure supplement(s) for figure 5:

**Figure supplement 1.** Genes at MAPL are preferentially expressed in roots.

# Discussion

This study profiled the rhizosphere inhabiting microbiota of several hundred maize genotypes under agronomically relevant field conditions. Through a 16 S rDNA-sequencing based approach, we identified a set of 150 rhizobiome traits based on 3626 ASVs that were highly abundant and consistently reproducible in this maize diversity panel. The phylogenetic tree in *Figure 1B* may deviate from the consensus microbial phylogeny since only the 350 bp ribosomal V4 region was used to establish the relationship between groups, and more accurate phylogenetic clustering should be considered in future studies with emphasis on the evolution of plant-microbe associations. In total, 79 out of the 150 rhizobiome traits (52%) showed significant evidence of being influenced by host plant genotype in one or more environmental conditions. The estimated heritability of rhizobiome traits in this study ranged from 0 to 0.757 for the +N treatment (mean 0.320) and from 0 to 0.839 for the -N treatment (mean 0.352). A comparable study of the rhizobiomes in a sorghum diversity panel estimated similar values (*Deng et al., 2021*). A previous study on the same maize diversity panel (*Wallace et al., 2018*) investigated the heritability of 185 individual OTUs and 196 higher taxonomic units measured in the leaf microbiome. The study reported only two OTUs and three higher taxonomic groups showing significant heritability using the same permutation test we employed in this study. This may indicate that plant genetics have a stronger influence on rhizosphere colonizing microbes than on leaf colonizing microbes. One reason for this may be that there is a direct pathway for plant-to-microbe communication via root exudates (*Doornbos et al., 2012*). In contrast, no equivalent exchange of chemical information has been reported above ground, with the possible exception of aerial root mucilage (*Van Deynze et al., 2018*).

We observed relatively higher heritability for rhizobiome traits quantified from plants grown in the -N treatment than under the +N treatment. This outcome is consistent with a model where the partnerships between microbiomes and plants were established in natural and early agricultural systems which were predominantly N limited (*Brisson et al., 2019*). N insufficiency in maize induces dramatic changes in physiology (*Ciampitti et al., 2013*), gene expression (*Chen et al., 2011*; *Singh et al., 2013*), root architecture (*Gaudin et al., 2011*), and root exudation (*Baudoin et al., 2003*; *Haase et al., 2007*; *Zhu et al., 2016*). Consistent with this, N fertilization or the lack thereof has substantial consequences on plant-microbe associations. In this study, 12% of rhizobiome traits were only significantly heritable under the +N treatment, and 18% only under the -N condition, and GWAS revealed plant-microbe associations at different genomic loci depending on the N treatment. Previous observations have also reported that rhizosphere microbial communities are highly sensitive to environmental conditions, in particular to N deficiency (*Meier et al., 2021*; *Zhu et al., 2016*). This finding emphasizes the need to optimize microbial communities not only for a specific host but also for specific levels of N fertilization.

Our results suggest that the capacity of maize plants to encourage or discourage colonization of the rhizosphere by specific microbiota has been a target of selection. The BayesS method leverages the relationship between the variance of SNP effects and MAF as a proxy of natural selection in the distant past. This method detects signatures of natural selection on SNPs associated with microbiome traits but is not directly indicative of selection acting on the particular microbes. Indeed, we observed purifying selection acting on genetic variants associated with the abundance of nine rhizosphere traits, 7 in the +N and 7in the -N environment, respectively. Several rhizosphere denizens whose abundance showed evidence of being a target of purifying selection in the host genome have been linked to plant growth promoting activities, most notably *Pseudomonas* (*Oteino et al., 2015*; *Preston, 2004*) and *Burkholderia* (*Bernabeu et al., 2015*; *Kurepin et al., 2015*). *Bacillus fumarioli,* which showed evidence of positive selection, has previously been observed in plant rhizospheres, particularly in maize (*Garbeva et al., 2008*), and several strains of *Bacillus* plant growth promoting activities (*Kumar et al., 2012*). Notably, not all traits that are heritable are expected to be under selection, as traits can be heritable, that is transmitted from one generation to the next, without impacting the fitness or performance of offspring individuals under the conditions under which recent natural and/or artificial selection has occurred. Additional functional analyses (i.e. inoculation experiments) are warranted to further approve the beneficial effects of the microbes on plant fitness, and to investigate how naturally occurring microbe-plant symbiosis may be harnessed for further crop improvement.

Among the 150 rhizobiome traits analyzed here, 62 showed a significant correlation with measurements of canopy coverage collected from the same field experiment. In particular, the observed link

between heritability of microbes and correlation with plant performance may indicate a symbiotic relationship of the host plant and root-associated microbes. However, while our data show correlations between microbe abundance and plant phenotypes, further experiments are required to determine the direction of causation and investigate potential mechanisms by which microbe abundance could influence phenotypic changes in the host. We observe that the majority of rhizobiome traits that are correlated with canopy coverage are both heritable and associated with one or more microbe-associated plant loci (MAPLs), and genes linked to variation in rhizobiome traits via GWAS were highly expressed in roots across genotypes in multiple independent gene expression datasets. This suggests a number of potential mechanisms for host plant genotypes to influence the composition of the rhizobiome.

For example, two of the three genes associated with the MAPL highlighted in *Figure 5* (Zm0001d023838 and Zm0001d023839) are preferentially expressed in roots (*Figure 5—figure supplement 1*). According to MaizeGDB, both are protein coding genes that have not yet been characterized in maize. Known Zm0001d023838 orthologs in *Arabidopsis* encode AUXILIN-LIKE1 and AUXILIN-LIKE2, and overexpression of auxilin-like proteins in *Arabidopsis* has been shown to inhibit endocytosis in root hair cells (*Ezaki et al., 2007*). Overexpression of auxilin-like proteins has also been shown to confer resistance to root-borne bacterial pathogens in rice (*Park et al., 2017*). This indicates a possible link between root hair physiology and an altered microbiome. Although substantial further experimentation and study remains necessary, adjusting the expression of particular MAPL genes identified here may be an avenue to directly influence and engineer the abundance of targeted microbial groups in the rhizosphere to the benefit of the plant.

We evaluated associations between rhizobiome traits and a number of whole plant phenotypes. The Buckler-Goodman maize diversity panel has been and continues to be utilized in field experiments to determine the genetic basis of many phenotypes across diverse environments. The datasets generated here link the abundance of 150 microbial groups in the rhizosphere to genetic variation in 230 genotypes across two N treatments. Combining these public datasets with plant phenotypes collected from the same genotypes in additional environments may lead to the identification of other cases where MAPLs are associated with variation in plant phenotypes or plant performance. The results presented in this study add to an increasing body of evidence that microbial communities are actively and dynamically shaped by host plant genetic variation and may serve as the foundation for future research into particular plant-microbe relationships that may be harnessed to sustainably increase crop productivity and resilience to abiotic stress.

## Materials and methods
### Field and experimental design
In this study, 230 maize (*Zea mays* subsp. *mays*) lines from the maize diversity panel (*Flint-Garcia et al., 2005*) were planted in May of 2018 and 2019 in a rain-fed experimental field site at the University of Nebraska-Lincoln's Havelock Farm (N 40.853, W 96.611). In both years, the experiment followed commercial maize. Individual entries consisted of 2 row, 5.3 m long plots with 0.75 m alleyways between sequential plots, 75 cm spacing between rows, and 15 cm spacing between sequential plants. In each year, the experimental field was divided into 4 quadrants and the complete set of genotypes was planted in each quadrant following an incomplete block design (**Supplementary Methods**, *Appendix 1—figure 1*). N fertilizer (urea) was applied at the rate of 168 kg/ha to two diagonally opposed quadrants before planting, while two quadrants were left unfertilized (-N treatment).

### Rhizobiome sample preparation and sequencing
In 2018, n=304 rhizosphere samples were collected from 28 maize genotypes (2 samples per subplot, 2 replicated plots per genotype and N treatment); and in 2019, n=3009 samples were collected from 230 genotypes (3 samples per subplot, 2 replicated plots per genotype and N treatment), listed in *Supplementary file 1*. Eight weeks after planting (2018: July 10 and 11; 2019: July 30, 31 and August 1), plant roots were dug up to a depth of 30 cm and rootstocks were manually shaken to remove and discard loosely adherent bulk soil. For each plant, all roots thus exposed were cut into 5 cm pieces and homogenized, and 20–30 ml randomly selected root material (with adherent rhizosphere soil) was collected to generate the rhizosphere samples (**Supplementary Methods**). DNA was isolated using

the MagAttract PowerSoil DNA KF Kit (Qiagen, Hilden, Germany) and the KingFisher Flex Purification System (Thermo Fisher, Waltham, MA, USA). DNA sequencing was performed using the Illumina MiSeq platform at the University of Minnesota Genomics Center (Minneapolis, MN, USA). In brief, 2 × 350 bp stretches of 16 S rDNA spanning the V4 region were amplified using V4_515 F_Nextera and V4_806 R_Nextera primers, and the sequencing library was prepared as described by Gohl (*Gohl et al., 2016*).

### Raw read processing and construction of microbiome dataset

Paired-end 16 S sequencing reads from 3,313 samples were processed in R 3.5.2 using the workflow described by Callahan (*Callahan et al., 2016a*), which employs the package dada2 1.10.1 (*Callahan et al., 2016b*). Taxonomy was assigned to amplicon sequence variants (ASVs) using the SILVA database version 138 (*Yilmaz et al., 2014*) as the reference. Raw ASV reads were subjected to a series of filters to produce a final ASV table with biologically relevant and reproducible 16 S sequences (*Supplementary file 1*). For the constrained ordination (CAP) analysis performed here, the weighted Unifrac distance metric was used with model distance ~year + genotype +nitrogen + block +sp + spb. Only ASVs that were highly abundant and repeatedly observed in both years of sampling were considered for downstream analysis. ASVs were clustered into 150 groups of rhizosphere microbes at the family, genus, and species level based on 16 S sequence similarity and the response of individual ASVs to experimental factors (see supplementary methods).

### Heritability estimation

Heritability ($h^2$) of rhizobiome traits was calculated separately for +N and -N conditions using maize genotypes in the 2019 dataset for which balanced data was available. For each of the 150 rhizobiome traits, combined ASV counts were normalized by converting to relative abundance and subsequent natural log transformation. Using these transformed values, $h^2$ was estimated following *Deng et al., 2021* for each rhizobiome trait using R package sommer 4.1.0 (*Covarrubias-Pazaran, 2016*). In short, $h^2$ is the amount of variance explained by the genotype term ($V_{genotype}$) divided by the variance of the genotype and the error ($V_{genotype}$ +$V_{error}$/n), where n=6 is the total number of samples (i.e., 2 replicates x 3 samples per replicate) used in this dataset. Heritability was tested for significance using a permutation test. For each trait, the genotype labels of microbial abundance data were shuffled 1000 times, and the distribution of heritabilities calculated from these shuffled datasets were used to assess the likelihood of observing the heritabilities calculated from the correctly labeled data under a null hypothesis of no host genetic control.

### Calculation of selection differentials and estimation of genetic architecture parameters

We estimated the fitness function relating the fitness-related trait, that is canopy coverage collected on August 22 (see section "Phenotyping of plant traits"), to the abundance of the microbial groups with a generalized additive model (GAM). To reduce biases due to environmental covariances (*Rausher, 1992*), we employed the BLUP values for both the rhizobiome traits and the fitness-related trait. Then, we obtained linear and quadratic selection differentials from the fitted GAM models using an R package (*Morrissey and Sakrejda, 2013*). We ran a total of 300 univariate models (150 microbial groups x 2 N treatments).

For the rhizobiome traits, a Bayesian-based method (*Zeng et al., 2018*) was used to estimate genetic architecture parameters simultaneously, including polygenicity (i.e. proportion of SNPs with non-zero effects), SNP effects, and the relationship between SNP effect size and minor allele frequency. For the analysis, genotypic data of the maize diversity panel was obtained from the Panzea database and uplifted to the B73_refgen_v4 (*Bukowski et al., 2018*; *Woodhouse et al., 2021*). To account for SNP linkage disequilibrium (LD), a set of 834,975 independent SNPs (MAF ≥ 0.01) were retained by pruning SNPs in LD (window size 100 kb, step size 100 SNPs, $r^2$ ≥0.1) using the PLINK1.9 software (*Chang et al., 2015*). In the analysis, the 'BayesS' method was used with a chain length of 410,000 and the first 10,000 iterations as burnin.

### Genome-wide association study

We chose to use the best linear unbiased prediction (BLUP) of the natural log transformed relative abundance of ASV counts as the dependent variable for the GWAS analysis. Since only a fraction of

genotypes were sampled from the 2018 field experiment, only sample data collected in 2019 was used for the BLUP calculation. A BLUP value was calculated for each microbial group and each treatment using R package lme4 (**Bates et al., 2015**). In the analysis, the following model was fitted to the data: Y ~ (1|genotype) + (1|block) + (1|split plot) + (1|split plot block)+error, where Y represents a rhizobiome trait (ln(ASV count of a microbial group / total ASV count in sample)) (**Supplementary Methods**, *Appendix 1—figure 1*). GWAS was performed separately for each rhizobiome trait and for both the +N and -N treatment using GEMMA 0.98 (**Zhou and Stephens, 2012**) with a set of 21,714,057 SNPs (MAF ≥ 0.05) (**Bukowski et al., 2018**). In the GWAS model, the first three principal components and the kinship matrices were fitted to control for the population structure and genetic relatedness, respectively. To mitigate false discoveries of GWAS, Bonferroni corrections were applied based on the effective number of independent SNPs (or effective SNP number) (**Li et al., 2012**). The effective SNP number for the genetic marker set and population employed in this study was determined to be N=769,690 independent markers as described previously (**Rodene et al., 2022**). Using an alpha value of 0.05, we determined a significance threshold of -log10(0.05/769,690)=7.2.

### RNA sequence analysis

Gene expression was analyzed using two independent datasets. The first dataset was obtained from Kremling (**Kremling et al., 2018**) and included RNA sequencing data from 7 different maize tissue types. The second RNA sequencing dataset was generated from root and leaf tissue collected 14 days after germination from both +N and -N treated pots using 4 genotypes from the maize diversity panel. Libraries were sequenced using the Illumina Novaseq 6,000 platform with 150 bp paired-end reads. Sequencing reads were mapped to the B73 reference genome (AGPv4) (**Jiao et al., 2017**; **Schnable et al., 2009**) and gene expression was quantified using Rsubread (**Liao et al., 2019**).

### Phenotyping of plant traits

A total of 17 plant traits were measured in the 2019 field experiment. First, 15 leaf physiological traits were measured on the same days the rhizobiome samples were collected, and included leaf area (LA), chlorophyll content (CHL), dry weight (DW), fresh weight (FW), as well as concentrations of the elements B, Ca, Cu, Fe, K, Mg, Mn, N, P, S, and Zn. Measurement of the leaf traits was carried out as previously described (**Ge et al., 2019**). Two aerial imaging traits, canopy coverage (CC) and excess green index (ExG), were collected on August 12, 2019, 11–13 days after rhizobiome sample collection (**Rodene et al., 2022**).

### Availability of data and materials

The sequencing data reported in this publication (3313samples) can be accessed via the following five Sequence Read Archive (SRA) accession numbers: PRJNA771710, PRJNA771712, PRJNA771711, PRJNA685208, PRJNA685228 (summarized under the umbrella BioProject PRJNA772177). Scripts used to analyze the data are available on GitHub (https://github.com/jyanglab/Maize_Rhizobiome_2022; **Rhizobiome, 2022**).

### Acknowledgements

This study is supported by National Science Foundation EPSCoR Cooperative Agreement OIA-1557417. In Memory of James R Alfano we thank him for his initiative and leadership at the Center for Root and Rhizobiome Innovation (CRRI). We also thank Edgar Cahoon and the CRRI team for setting up and maintaining an exemplary collaborative environment. We further acknowledge Tom Clemente, Karin van Dijk, Daniel Schachtman, Ellen Marsh, Lisa Vonfeldt, Alan Muthersbaugh, Jenny Stebbing and TJ McAndrew, for technical support. Lastly, we thank the many scientists who assisted in collecting rhizosphere samples: Laure-Olivia Mbouang Angoua, Bryce Askey, Abbas Atefi, Natalie Belz, Erin Bertone, Eledon Beyene, Alexandra Bradley, Amanda Butera, Christian Butera, Madelyn Calvert, Noah Carroll, Jessica Chen, Sierra Conway, Floreana Cordova, Xiuru Dai, Semra Palali Delen, Yavuz Delen, Tessa Durham-Brooks, Samuel Eastman, Alex Enerson, Ashley Foltz, Nick Friedman, Cierra Goerish, Wihib Hankore, Davron Hanley, Aris Hino, Chun Yin Ho, J Preston Hurst, Kylie Irene, Panya Kim, Nataniel Korth, Courtney Krsnak, Enzo Lamontia, Zhikai Liang, Xiangdong Liu, Angelique Malcolm, Rajan Mediratta, Chenyong Miao, Xiaoxi Meng, Levi Nigro, Alejandro Pages, Connor Pedersen, Nathaniel Pester, Sam Polk, Raghuprakash Kastoori Ramamurthy, Eric Rodene, Daniel

Santano de Carvalho, Emma Sheridan, Aris Shino, Isabel Sigmon, Taylor Stratman, Guangchao Sun, Michael Tross, Misty Wehling, Florian Wurtele and Zhikai Yang.

## Additional information

### Funding

| Funder | Grant reference number | Author |
|---|---|---|
| National Science Foundation | Cooperative Agreement OIA-1557417 | Jinliang Yang |

The funders had no role in study design, data collection and interpretation, or the decision to submit the work for publication.

### Author contributions

Michael A Meier, Data curation, Formal analysis, Validation, Investigation, Visualization, Methodology, Writing - original draft; Gen Xu, Data curation, Formal analysis; Martha G Lopez-Guerrero, Guangyong Li, Christine Smith, Brandi Sigmon, Joshua R Herr, Data curation; James R Alfano, Conceptualization, Data curation, Funding acquisition; Yufeng Ge, Data curation, Project administration; James C Schnable, Conceptualization, Resources, Data curation, Funding acquisition, Project administration, Writing - review and editing; Jinliang Yang, Resources, Data curation, Formal analysis, Supervision, Funding acquisition, Investigation, Visualization, Methodology, Project administration, Writing - review and editing

### Author ORCIDs

Michael A Meier http://orcid.org/0000-0002-7727-6561
Joshua R Herr http://orcid.org/0000-0003-3425-292X
Jinliang Yang http://orcid.org/0000-0002-0999-3518

### Decision letter and Author response

Decision letter https://doi.org/10.7554/eLife.75790.sa1
Author response https://doi.org/10.7554/eLife.75790.sa2

## Additional files

### Supplementary files

• Supplementary file 1. Feature table (3,306 samples by 3,626 ASVs) from which our results were generated, alongside the sample metadata collected in this study.

• Supplementary file 2. Taxonomically annotated list of 3,626 16 S sequences that comprise the core maize microbiome used for this analysis and may serve as a reference to identify the same maize-associated ASVs in future experiments.

• Supplementary file 3. List of the 150 microbial groups defined in this study alongside relevant summary statistics, such as abundance, heritability, selection coefficients, and correlations with plant traits under both N treatments.

• Supplementary file 4. List of 229 Buckler-Goodman maize genotypes with the corresponding measurements of all 17 plant and 150 rhizobiome traits analyzed here under both N treatments. Sample-level data is published for aerial imaging (*Rodene et al., 2022*).

• Supplementary file 5. List of 622 plant loci (10 kb genomic regions) that exhibit significant association with one or more microbial groups, including the IDs of nearby (+/-10 kb) genes.

• Supplementary file 6. Plots of phylogeny, abundance and response to N treatment for all microbial families present in this dataset, with clustering of ASVs into the microbial groups used here.

• Transparent reporting form

### Data availability

All data generated or analysed during this study are included in the manuscript and supporting file.

The following dataset was generated:

| Author(s) | Year | Dataset title | Dataset URL | Database and Identifier |
|---|---|---|---|---|
| Meier MA, Schnable JC, Yang J | 2021 | Rhizosphere microbiome of 230 maize genotypes under standard and low nitrogen treatment | https://www.ncbi.nlm.nih.gov/bioproject/PRJNA772177 | NCBI BioProject, PRJNA772177 |

The following previously published datasets were used:

| Author(s) | Year | Dataset title | Dataset URL | Database and Identifier |
|---|---|---|---|---|
| Bukowski R, Guo X, Lu Y, Zou C, He B, Rong Z, Wang B, Xu D, Yang B, Xie C, Fan L | 2018 | Construction of the third-generation Zea mays haplotype map | https://www.ncbi.nlm.nih.gov/bioproject/PRJNA389800 | NCBI BioProject, PRJNA389800 |
| Kremling KA, Chen SY, Mh SU, Lepak NK, Romay MC, Swarts KL, Lu F, Lorant A, Bradbury PJ, Buckler ES | 2018 | Dysregulation of expression correlates with rare-allele burden and fitness loss in maize | https://www.ncbi.nlm.nih.gov/bioproject/?term=PRJNA383416 | NCBI BioProject, PRJNA383416 |

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

## Appendix 1

## Supplementary Methods
### Field and experimental Design
The experimental field was divided into 4 quadrants, which were separated and surrounded by a buffer of an industrial hybrid genotype (B73xMo17) (*Appendix 1—figure 1*). The complete set of genotypes was planted in each quadrant where possible. Each quadrant was in turn divided into 4 split plots and a subset of the maize association panel was randomly assigned to each split plot based on the distributions of flowering time and plant height. Phenotypes were divided at the median value to create 4 flowering time / height categories: early/tall, late/tall, early/short, and late/short. Each split plot was further divided into 3 split plot blocks, and each split plot block was divided into 21 subplots in 3 ranges and 7 columns. Thus 252 subplots were available in each quadrant of the field. In each of 12 split plot blocks per quadrant, a t least one subplot was randomly selected and assigned the hybrid genotype (B73xMo17) to be used as a check to test for differences between geographical field locations. two check genotypes (B73xMo17 and B37xMo17) were used in 2018, and a single check genotype (B73xMo17) was used in 2019. Plant growth across the field was determined uniform within quadrants using the checks as reported in a sister study on the same experimental field (*Rodene et al., 2022*). Any subplots across the field that remained empty due to seed unavailability were filled with the check genotype as well.

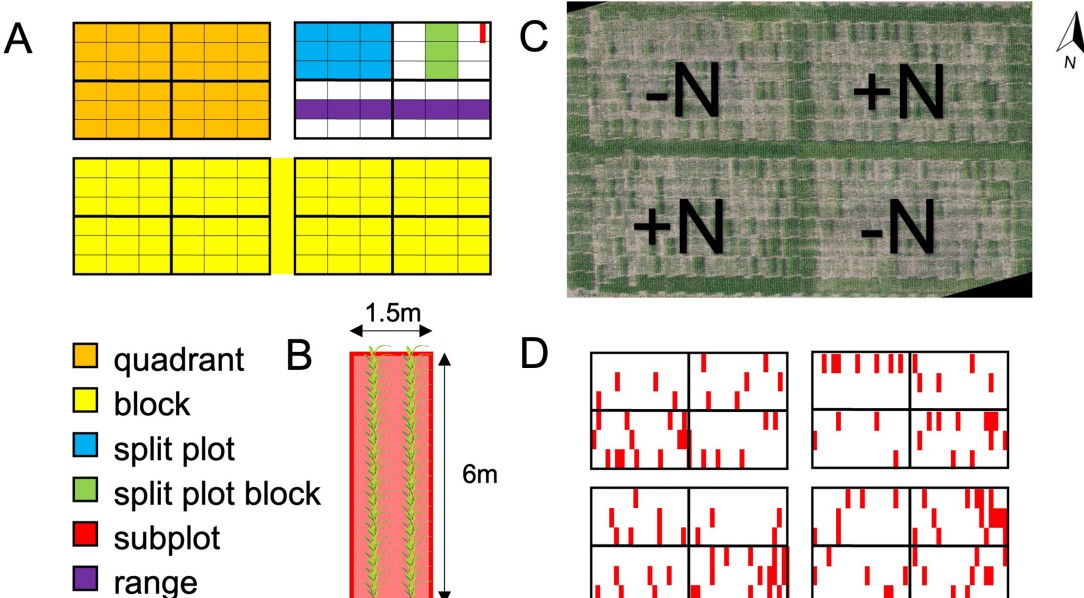

**Appendix 1—figure 1.** Field experimental design. (**A**) Up to 230 maize genotypes were represented in each of 4 quadrants in 2 replicate blocks. Quadrants were planted in 6 ranges and divided into 4 split plots. Each split plot was divided into 3 split plot blocks, and each split plot block was divided into 21 subplots for a total of 252 subplots per quadrant. (**B**) Each 1.5 m (5 ft) x 6 m (20 ft) subplot (experimental unit) consisted of two rows of 36 maize plants of the same genotype, with a spacing of 75 cm (30 in) between rows and 15 cm (6 in) between plants. (**C**) Photomosaic of the 2019 field at flowering time. N fertilizer was applied to the NE and SW quadrants before planting. (**D**) 128 subplots across the field (marked in red) were planted with a check genotype (B73xMo17) in order to be able to quantify and control for spatial variation.

In 2018, dry N fertilizer (urea) was applied to two diagonally opposed quadrants before planting at the rate of 140 kg/ha (+N treatment) while two quadrants were left unfertilized (-N treatment). In 2019, liquid N fertilizer (urea) was applied at the rate of 168 kg/ha. Both N treatments were thus represented in a northern block (NW and NE quadrants) and in a southern block (SW and SE quadrant). We assigned the blocks this way because of a 3 m increase in elevation from the north end of the field to the south end.

## Rhizobiome sample preparation and sequencing

In 2018, rhizosphere samples were collected from 28 genotypes. These include, B73, the roothairless3 mutant of B73 (*Park et al., 2017*), two check hybrids (B73xMo17 and B37xMo17) and a subset of the Buckler-Goodman panel including 16 parent lines of the nested association mapping population (NAM) described by *McMullen et al., 2009*. 8 weeks after planting, 2 subsamples per genotype were collected per quadrant and 12 subsamples for checks, where each subsample was taken from the combined root material of two adjacent plants. This resulted in a total of 26*4*2+2*4*12=304 samples. In 2019, rhizosphere samples were collected in triplicates from all 1,008 subplots within 3 days, 8 weeks after planting, when the majority of plants had reached the tasseling stage. One of the two rows in each subplot was randomly selected, and 3 individual randomly selected plants within the row (subsamples) were sacrificed for rhizosphere collection. As a small fraction of subplots had poor germination and/or no surviving plants on the day of sampling, the final number of rhizosphere samples collected was 3,009. Rhizosphere samples were placed on ice immediately after collection and shipped to the lab to be processed on the same day.

To wash the tightly adherent rhizosphere soil layer off the roots, tubes were filled up to the 40 ml mark with autoclaved PBS buffer (46 mM NaH2PO4, 60 mM Na2HPO4, 0.02% Silwet-77), and shaken horizontally at 8000 rpm for 30 s. Rhizosphere suspension was filtered through a 100 µm nylon cell strainer (Celltreat Scientific Products, Pepperell, MA, USA) into a fresh 50 ml tube to capture root debris and large soil particles. Rhizosphere samples were frozen in suspension at –20 °C until further processing. DNA was isolated from rhizosphere soil using the MagAttract PowerSoil DNA KF Kit (Qiagen, Hilden, Germany) and purified using the KingFisher Flex Purification System (Thermo Fisher, Waltham, MA, USA) with minor modifications to the protocol: Rhizosphere samples that were kept in suspension were thawed on ice, pelleted soil was resuspended by inverting tubes, and 500 µl soil suspension was added to the 96-well sample plates. To avoid cross contamination of wells during pipetting, plates were sealed beforehand with parafilm and the cover was pierced with the pipette tip to transfer the rhizosphere suspension into the intended well. Two plates were prepared at a time and centrifuged for 10 min at 4000 x g to pellet soil. Supernatant was carefully removed with a multichannel pipette and 96-well plates with approximately 100–250 mg rhizosphere soil per well were frozen at –20 °C until further processing. On the day of DNA isolation, the bead mill substrate was added to the frozen soil pellets, soil was thawed on ice and the remainder of the protocol was followed as per the manufacturer's instructions. We recommend this modified procedure for large numbers of samples as it is cleaner, faster, and better reproducible than scooping soil from pellets in sample tubes. Concentration of isolated DNA was measured fluorometrically with the QuantiFluor dsDNA System (Promega, Madison, WI, USA) as per the manufacturer's instructions. DNA isolation was repeated for any samples that failed to reach a concentration of at least 1 ng/µl.

A 350 bp stretch of 16 S rDNA spanning the V4 region was amplified using V4_515 F_Nextera ( TCGTCGGCAGCGTCAGATGTGTATAAGAGACAGGTGCCAGCMGCCGCGGTAA) and V4_806 R_ Nextera (GTCTCGTGGGCTCGGAGATGTGTATAAGAGACAGGGACTACHVGGGTWTCTAAT) primers on several Illumina MiSeq runs. Oligonucleotide PCR blockers (PNA Bio INC, Thousand Oaks, CA, USA) targeting mitochondrial and chloroplast sequences were applied in the primary V4 amplification to reduce amplification of templates derived from the plant host. Up to 128 barcoded samples were pooled per sequencing run. In total, 304 samples in 2018 and 3009 samples in 2019 were sequenced on the same Illumina MiSeq machine.

## Raw read processing and construction of microbiome dataset

Cluster computing resources at the UNL Holland Computing Center were used for computationally demanding steps. To construct the microbiome dataset, 350 bp raw sequencing reads were trimmed using filterAndTrim() at 240 bp (forward reads) and 200 bp (reverse reads), respectively. Amplicon sequence variants (ASVs) were inferred using dada() and forward and reverse reads were merged with mergePairs(). A sequence table was generated using makeSequenceTable() and chimaeras were removed using removeBimeraDenovo(). Taxonomy was assigned to ASVs with assignTaxonomy() using the SILVA database version 138 (*Yilmaz et al., 2014*) as a reference. SILVA was our taxonomy of choice because it is a relatively large 16 S sequence database compared to alternative databases, it is regularly maintained and updated and it is widely used in ecological research, making our results comparable to other 16 S studies. (*Balvočiūtė and Huson, 2017*). Taxonomic training data formatted for DADA2 (silva_nr99_v138_wSpecies_train_set.fa.gz) was obtained from https://zenodo.org/

record/3986799#.X3zmypNKh24, as referenced by https://benjjneb.github.io/dada2/training.html on GitHub. 16 S reads and sample data were prepared in an R Phyloseq object for further processing.

Raw ASV reads were subjected to a series of filters to produce a final ASV table with biologically relevant 16 S sequences:

1. Removed chimaeric 16 S reads using removeBimeraDenovo()
2. Removed sequences with <20 total observations
3. Removed sequences that did not map to either Bacteria or Archaea
4. Removed chloroplast sequences
5. Removed mitochondrial sequences
6. Removed ASVs that were not observed in at least 5% (166) of all samples
7. Removed ASVs that were not observed in both years 2018 and 2019
8. Removed 53 out of 160 genera and families that had fewer than 5 unique ASVs and 7 samples with <100 ASV counts

Step 6 resulted in 4,632 common ASVs that were detected in at least 5% of the samples, representing 120,004,239 of the raw reads. Constrained ordination and PERMANOVA analyses of the 4,632 ASVs identified a strong effect of N treatment as well as other experimental factors on ASV abundance (*Appendix 1—figure 2*). This observation is consistent with previous observations that environmental factors play an important role in determining the composition of the root associated microbiome diversity (*Floc'h et al., 2020*; *Meier et al., 2021*; *Schlatter et al., 2020*). Of the 4,632 common ASVs, 3,728 (or 80.5%) were highly abundant and observed in samples collected from both the 2018 and 2019 growing seasons (step 7). Removing ASVs that could not be repeatedly observed in multiple years reduced the complexity of the data set by 19.5% at the cost of a 2.3% loss in diversity (Shannon diversity reduced from 6.4 to 6.3, *Appendix 1—figure 2—Figure supplement 1*). Finally, removing taxa (genus or family) with less than 5 observed ASVs yielded a dataset of 3,626 ASVs, 3,306 samples, and 105,745,986 total ASV counts. This final core microbiome encompasses <1% of initial ASVs and ~50% of initial observations. The ASV table from step 8 was converted to relative abundances and values were transformed with the natural logarithm. A phylogenetic tree was constructed from the final set of 3,626 ASVs using mafft v. 7.404 (*Katoh and Standley, 2013*) for multiple alignment and fasttree v. 2.1 (*Price et al., 2010*) and the phylogenetic tree was attached to the phyloseq object and plotted using the ggtree R package (*Yu, 2020*).

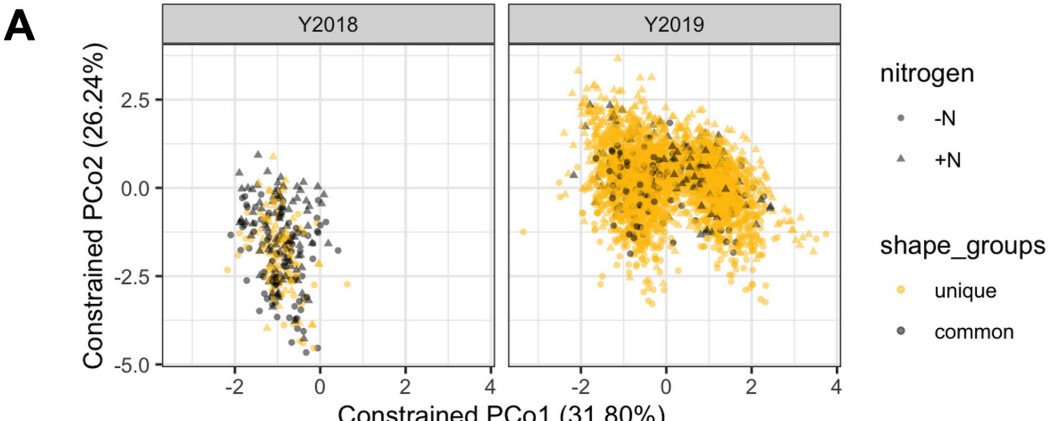

**Appendix 1—figure 2.** PERMANOVA results. It was calculated from the log(relative abundance) of 4,632 ASVs. Each dot represents a sample. Genotypes common to 2018 and 2019 panel are marked in grey.

The online version of this article includes the following figure supplement(s) for appendix 1—figure 2:

**Appendix 1—Figure 2 supplement 1.** Retaining ASVs observed in both years reduces dataset complexity with minimal loss of diversity.

## Clustering of ASVs into microbial groups

ASVs were clustered into groups of rhizosphere microbes at the family, genus, and species level using a procedure described previously (*Meier et al., 2021*). First, the 3,626 ASVs in the present study were grouped at the family level (the lowest taxonomic rank for which all ASVs were successfully annotated) and the phylogenetic tree derived from 16S V4 alignment was plotted alongside taxonomic annotation at the genus and species level. Because the ASVs are derived from short reads and may constitute variations not covered in the SILVA database, annotation at the genus and species level was often not possible. To close these gaps and form biologically meaningful groups of ASVs at low taxonomic ranks with better confidence, we examined the overall abundance of each ASV as well as the differential abundance in response to the N treatment alongside the sequence-based clustering. The premise here is that ASVs derived from biologically closely related individual microbes are similarly abundant in our dataset and respond similarly to the N treatment imposed on the field, in addition to similar 16 S sequences due to common ancestry. An example is given in *Appendix 1—figure 3* with a subset of ASVs assigned to the *Burkholderiaceae* family. The plots used to determine all 150 microbial groups in this study are available in *Supplementary file 6*.

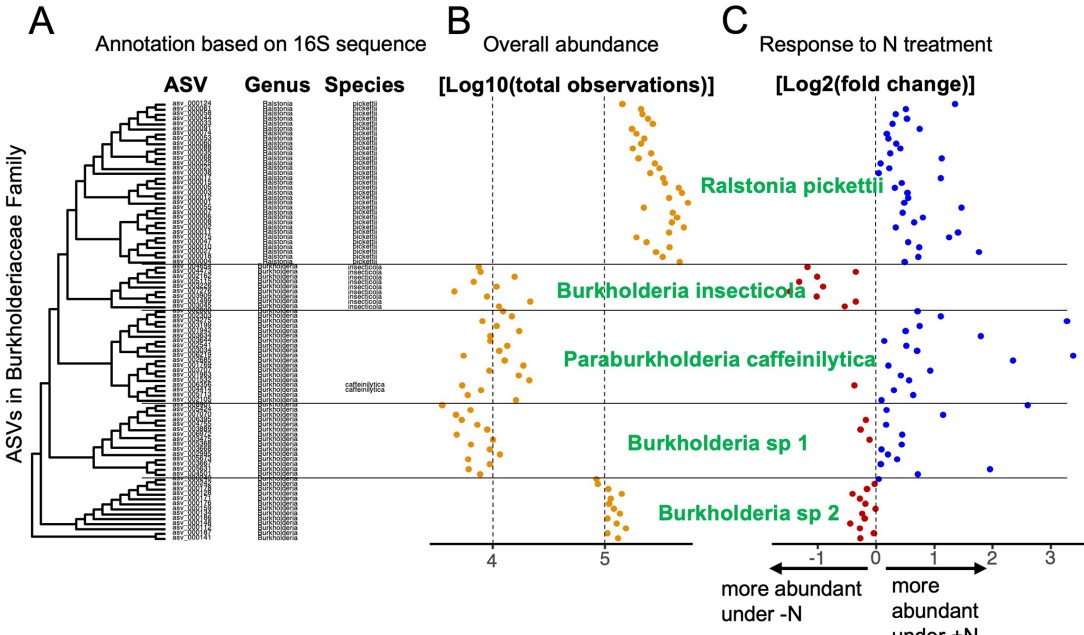

**Appendix 1—figure 3.** Microbial groups are derived from taxonomic data and experimental data. An example is given using a subset of the ASVs in the *Burkholderiaceae* family. (**A**) Phylogenetic clustering of ASVs based on 16S V4 alignment. ASVs are annotated at the genus and species level using the SILVA database. Note that for some ASVs, annotation at the species level is missing, although the phylogenetic tree suggests divergent groups at the species level. Overall abundance in the dataset (**B**) of each ASV and differential abundance in response to the N treatment (**C**) were used in tandem with sequence-based clustering to group ASVs with similar features into microbial groups at sub-genus resolution (labeled in green). In this example, the genus *Ralstonia* constitutes a monophyletic cluster of ASVs which were all successfully assigned to the species *R. pickettii* (**A**). This uniform group is also reflected in relatively uniform abundance (**B**) and positive response to N treatment (**C**). On the other hand, most ASVs in the *Burkholderia* genus could not be annotated at the species level, even though the phylogeny suggests at least 4 distinct groups below the genus level. The first group, *Burkholderia insecticola* was identified at the species level without fail and once again, this is reflected in uniform abundances as well as a consistently negative response to N treatment. Within the next cluster two ASVs are assigned to *Paraburkholderia caffeinilytica*, and we assigned all other ASVs in the same cluster to the same species because they showed consistent abundance and response to treatment. In the remaining two clusters, no ASVs could be annotated at the species level, hence we assigned a number to the unassigned species (*Burkholderia sp 1* and *sp 2*). Experimental data confirms that the two clusters should be treated as separate microbial groups because *Burkholderia sp 2* is roughly 10 times as abundant as *Burkholderia sp 1* and we observe opposite responses to N treatment.

## Heritability estimation

To calculate heritability (h2), read counts from 3 subsamples were pooled for each subplot. Combined counts were then normalized by converting to relative abundance and subsequent natural log transformation, which yielded a subplot-level measure of microbial abundance, replicated in 2 blocks. The following linear mixed model was used with all random effects: Y=genotype + block +error. Y is the log-transformed relative abundance of each microbial group in each subplot-level sample, the blocks and subplots are as outlined in (*Appendix 1—figure 1*). Heritability was tested for significance using a permutation test in which microbial abundance data for each trait was shuffled and heritability calculated anew 1,000 times. p-values indicating heritability were calculated by tallying the number of permutation h2 scores exceeding the observed h2 and dividing by the number of permutations. Traits with a *P*-value <0.05 were deemed "heritable" under either or both N treatments.

## Estimation of genetic architecture parameters

SNPs in high linkage disequilibrium (LD) were pruned using the "indep-pairwise" command of with a LD threshold of r2=0.1. In the GCTB analysis, the BayesS model was used with the chain length of 410,000 and burnin 10,000. One example command used for the GCTB analysis is "gctb –bfile

282_GCTB_G `--pheno` gctb_blup_stdN_150_tax_groups.txt `--mpheno` 28 `--out` Results_HN/asv_000013 `--bayes` S `--pi` 0.05 `--hsq` 0.5 S 0 `--wind` 0.1 `--chain-length` 410000 `--burn-in` 10000".

## Genome-wide association study

GWAS was performed using GEMMA 0.98 (*Zhou and Stephens, 2012*) with the following parameters: gemma-0.98 -bfile {snp_file} -k {kinship_matrix} -c {pca_file} -p {traits_file} -lmm 1 n {trait_num} -outdir {outdir_path} -o T{trait_num} -miss 0.9 r2 1 -hwe 0 -maf 0.01'. Blup values were summarized in a trait matrix (214 genotypes x 150 traits) for all 150 rhizobiome traits and for all 214 maize genotypes for which high quality SNP data was available. To conserve disk space, SNP information was only retained in each ASV if a response at p_wald $<10^{-2}$ was observed. To identify genomic loci with high counts of significant SNPs, the genome was split into bins of 10 kbp, and the number of significant SNP signals at a threshold of p_wald $<10^{-5}$ was counted for each bin.

