## [Editor Report]

It is widely assumed that plants actively manipulate their root associated microbiomes although the genetic factors that contribute to this process have yet to be discovered. The authors catalog the root-associated bacteria that associate with diverse maize lines, grown under low and high nitrogen treatments and through a genome-wide association study, identify candidate genetic loci that influence the plant associated microbiome. In addition to the interesting insights reported in the present manuscript, these data are likely to be used for and/or compared to, in many future studies.

---

## [Decision Letter]

**Decision letter after peer review:**

Thank you for submitting your article "Maize root-associated microbes likely under adaptive selection by the host to enhance phenotypic performance" for consideration by *eLife*. Your article has has been reviewed by three peer reviewers, including Rebacca Bart as the Reviewing Editor and Reviewer #1, and the evaluation has been overseen by Detlef Weigel as the Senior Editor.

The reviewers have discussed their reviews with one another, and we have drafted this to help you prepare a revised submission. Overall, we are all enthusiastic about this manuscript. We believe that *eLife* will be a good home for the paper, assuming you are able to address the below points.

Essential revisions:

1) Strengthen the data/analysis to support the claim in the title, abstract and text or revise the title to more accurately reflect the findings. We all felt that currently, the findings are overstated based on the presented data. As is, we feel that a more appropriate conclusion would be: "loci under selection in maize genome affect rhizosphere composition" or similar. We are currently not convinced that the microbes are in-fact driving selection, opposed to an indirect effect. Please include additional analysis to support this claim, or revise the text to avoid overstating and include discussion of this important caveat. Please see specific comments throughout the reviews.

2) Revise the text to more clearly describe the methods and logic behind the analyses. Please see specific comments throughout the reviews. Most importantly, first, we all feel that more care is needed in describing how the 150 microbial traits were defined. We did consider your previous paper, but that did not resolve our questions about how these specific 150 traits were defined from these specific data. Second, we have concern about FDR given the size of these datasets. Please revise the text to address these issues.

*Reviewer #1 (Recommendations for the authors):*

1. Please point to Support file 1 in line 143-146. I was confused about how this math worked out, until I look at the Supplementary file.

2. Lines 191-193 appear to be a different font color.

3. Please note the age of plants used for the previously available RNAseq data.

4. Supp Figure 5. Is PLAM supposed to be MAPL?

5. Consider revising lines 385-395 for clarity. I found this difficult to follow until I looked at Supplemental Figure 5. Given that this is a pretty important part of the paper, I think it should be understandable on its own (without needing to reference the supplemental figures). For example, from the text, it is not immediately obvious that the 30 traits that positively correlate with CC are from the 15 from +N and 29 from +N and -N.

6. You might also consider specifically referring to the microbe traits as 'microbe traits' and plant traits as 'plant traits' as I found this confusing in a few places.

7. Line 505: remove 'however'.

8. The final figure and analyses are exciting and very promising! Yet, you seem to shy away from emphasizing this point. I appreciate caution in drawing conclusions here but think that another sentence or two after line 431 would be appropriate.

*Reviewer #2 (Recommendations for the authors):*

This manuscript presents one of the largest 16S rRNA amplicon datasets I have ever seen. The study reports on a field experiment done with 230 maize genotypes under 2 nitrogen fertilization regimens. The abundance of each taxon within the root microbiota of over 3000 samples were used to perform two separate analyses: (i) a GWAS, using the abundance of each taxon as a quantitative trait and (ii) a phenotypic study, identifying microbial taxons correlated with different plant phenotypes (mainly canopy coverage).

This is an impressive manuscript which will help pave the way for a better understanding of heritability in the plant microbiota, which is a major open question in our field. In massive screening studies such as this one, it is often challenging to deliver a concise take-home message or a mechanistic insight. Here, the actual functions of "MAPL" genes are not looked at in detail, besides the looking at their expression profiles across the plants (which is an interesting analysis in of itself). I therefore urge the authors to make their data, in the form of supplementary tables, as accessible as possible for follow up studies on the loci that they detected.

From what I can understand, and from some consultation with colleagues, the GWAS and related analysis were performed rigorously and with the appropriate normalizations and controls. However, I stress that I am not an expert in population genetics.

Comments:

One methodological aspect is unclear to me. How were the 150 "rhizobiome traits" defined? This seems like a taxonomic classification, but I am not clear how, or why, this was done? Also, the term "rhizobiome traits" is somewhat confusing, especially since these 150 vectors are referred to by different names throughout the manuscript. This is a key part of the analysis and should be explained clearly.

A second and related comment, is why are the only "rhizobiome traits" that the authors consider taxon abundances? Other quantities can be extracted from this dataset: α diversity, β diversity (e.g the 1st and 2nd axes of the ordination). You could also try to extract a measure of absolute abundance from the ratio of bacterial to plastid/mitochondrial reads. These would provide a more holistic picture of how plant genotype influences the microbiome composition.

I am perceiving a bit of a disconnect between the GWAS and the phenotypic comparisons. Indeed, the authors show that heritability is correlated with the correlation with CC, but I would assume that much of the phenotypic differences observed in the first place are a result of the genotypic variability, but I do not see a unified model that accounts for the relationship between plant genotype, microbiome composition and plant phenotype. Am I missing something here?

The relative abundance and the prevalence of ASVs is essentially ignored, beyond the initial screening described in the appendix. It would be important to know for example if heritable taxa are also abundant ones?

Line 37: study, singular.

Line 250: could the results of the PERMANOVA from the supplementary figure be highlighted also in the text?

Line 253: I do not understand how these 150 groups were classified. Also, why do you say that they are functionally distinct? All you have is their 16S sequence, there is no functional information (see comment above).

Figure 1b: As far as I could tell, the phylogenetic reconstruction in this tree does not match the consensus phylogeny for bacteria. This is more likely an error in their tree building algorithm than in the consensus tree and should be corrected.

Line 275: This could be measured (pagel's λ for example). This also raises the question why were all the ASVs binned to 150 groups?

Line 285: you didn't really frame this as a hypothesis.

Line 291: please explain the S parameter.

Line 305: these 3 traits are not independent traits. More like 3 facets of the same trait.

Figure 3: why wasn't this analysis done for plant traits as well?

Line 505: "however" – I do not understand how this is contrary to the previous sentence.

The terms "rhizosphere", "rhizobiome" and "root associated microbiome" seem to be used interchangeably here. Please sort this out.

Line 779: was the DNA extraction kit substituted mid project? How did this affect the results?

*Reviewer #3 (Recommendations for the authors):*

Comparing patterns of selection in N+ and N- environments can be done using methods that have been long established in evolutionary genetics: for example, calculating genotypic selection gradients in each of the treatments (Rausher 1992, https://doi.org/10.1111/j.1558-5646.1992.tb02070.x)

In theory, the patterns of heritability in rhizobiome features could be compared between treatments using a paired t-test.

---

## [Author Response]

Essential revisions:1) Strengthen the data/analysis to support the claim in the title, abstract and text or revise the title to more accurately reflect the findings. We all felt that currently, the findings are overstated based on the presented data. As is, we feel that a more appropriate conclusion would be: "loci under selection in maize genome affect rhizosphere composition" or similar. We are currently not convinced that the microbes are in-fact driving selection, opposed to an indirect effect. Please include additional analysis to support this claim, or revise the text to avoid overstating and include discussion of this important caveat. Please see specific comments throughout the reviews.

Thanks for the suggestion. We agree with this point and revised the title and abstract to highlight that the results are largely driven by association analyses. Please see the revised abstract in the edited manuscript. The title has been changed to “Association analyses of host genetics, root-colonizing microbes, and plant phenotypes under different nitrogen conditions in maize”. And we also added additional discussions about the caveats of some of the analyses.

2) Revise the text to more clearly describe the methods and logic behind the analyses. Please see specific comments throughout the reviews. Most importantly, first, we all feel that more care is needed in describing how the 150 microbial traits were defined. We did consider your previous paper, but that did not resolve our questions about how these specific 150 traits were defined from these specific data.

We added a section “Clustering of ASVs into microbial groups” in the Materials and methods (See detailed responses for the reviewer #1). Additionally, we added a supplementary figure to better explain how microbial groups were clustered.

Second, we have concern about FDR given the size of these datasets. Please revise the text to address these issues.

We agree that FDR adjusted p-values allow us to identify associations between plant genomic loci and microbial groups with fewer false discoveries.

In our first submission we chose an arbitrary cutoff of -Log10(p) = 5 to distinguish significant vs. non-significant GWAS signals. To improve on this, p-values were adjusted based on the effective SNP number (Mural et al., 2021). The methods section was updated to reflect this.

Added Text:

“To mitigate false discoveries of GWAS, Bonferroni corrections were applied based on the effective number of independent SNPs (or effective SNP number) (Li et al., 2012). The effective SNP number for the genetic marker set and population employed in this study was determined to be N = 769,690 independent markers as described previously (54). Using an α value of = 0.05 we determined a significance threshold of -log10(0.05/769,690) = 7.2.”

Using the new significance threshold of 7.2, we re-calculated the number of significant GWAS signals across all microbial groups.

**Author response table 1. sa2table1:** 

Significance threshold	Total Significant SNPs across all 150 microbial groups (+N)	Total Significant SNPs across all 150 microbial groups (-N)
5 (previous)	46,830	43,252
7.2 (new)	624	465

With the more stringent criteria this yielded a greatly reduced number of 1,089 significant GWAS signals. Using this reduced set of GWAS signals we proceeded to re-evaluate 10kb genomic windows in which strong associations with one or more microbial groups are observed (microbe-associated plant loci, MAPLs).

To create the summary data for Figure 3A, our previous method of choice was to tally the number of SNPs above the significance threshold in every 10kb bin (Author response image 1). This approach is less reliable with the reduced set of 1,089 GWAS signals because on average, fewer SNPs are observed per bin. Thus, we opted to instead report the mean p-value of all significant SNPs in each 10kb genomic window (Author response image 1). Incidentally, this also eliminated the need to filter out the strongest signals using quantiles, which are arbitrary and difficult to interpret. Finally, to further reduce the risk of false discoveries, we only retained genomic windows that showed at least two significant GWAS signals (Author response image 1). Thus, we now report a set of 119 MAPLs, which are a subset of the 467 MAPLs previously reported. As pointed out in the plot, the MAPL on chromosome 10 that shows a strong association with the f_Comamonadaceae microbial group and was analyzed in detail in Figure 5 remains prominent in the FDR adjusted dataset.

**Author response image 1. sa2fig1:** Identification of microbe-associated plant loci (MAPLs), comparison of the previous approach (A) and more stringent approach with reduced false discovery rate (B, C). Data is only plotted for the -N treatment, counts of MAPLs are for both N treatments.

Figure 3A was updated using the new stringency criteria and the resulting set of 119 MAPLs. We further annotated the microbial group(s) associated with each MAPL on the maize genome. Likewise, gene expression was calculated anew for the updated set of 119 MAPLs in Figure 3B and C and the same patterns were observed as before.

Dataset 5 was updated to include all 622 microbe-associated plant loci (MAPLs) that contain at least one significant association with any of the 150 microbial groups.

Reviewer #1 (Recommendations for the authors):1. Please point to Support file 1 in line 143-146. I was confused about how this math worked out, until I look at the Supplementary file.

Thanks. We added the reference to Supplementary File 1.

2. Lines 191-193 appear to be a different font color.

We fixed this in the revised manuscript.

3. Please note the age of plants used for the previously available RNAseq data.

We specified the plant tissues were collected during germination and at flowering time.

4. Supp Figure 5. Is PLAM supposed to be MAPL?

PLAM represents plant locus-associated microbe. We removed this term to avoid confusion with MAPL (microbe-associated plant locus).

5. Consider revising lines 385-395 for clarity. I found this difficult to follow until I looked at Supplemental Figure 5. Given that this is a pretty important part of the paper, I think it should be understandable on its own (without needing to reference the supplemental figures). For example, from the text, it is not immediately obvious that the 30 traits that positively correlate with CC are from the 15 from +N and 29 from +N and -N.

We thank the reviewer for the feedback. To clarify the relevant section, we revised the text. In addition, positive and negatively correlated microbial groups were marked in Figure 4. See revised text below.

“In total, 62 microbial groups – more than expected by chance (permutation test p < 0.001) – were, either positively or negatively correlated with CC (Figure 4). 15 microbial groups were associated with CC under +N treatment, 18 under -N treatment, and 29 showed a significant association under both N treatments (Supplementary Figure 5). 30 traits under +N and 35 under -N were positively correlated with CC. 14 traits under +N and 12 under -N were negatively correlated with CC. Among the same microbial groups, 44/62 (71%) showed significant evidence of host plant genetic control under either or both N treatments and 56/62 (90%) are associated with 255/395 (65%) genes in 174/467 MAPLs (39%) identified across the maize genome (Supplementary Figure 5). Under both N treatments, we observe an association between heritability and the correlation with CC, which was statistically significant (r = 0.39, p = 4x10-6) for +N and even more significant (r = 0.49, p = 1.7x10-9) under the -N regime (Figure 4B).”

6. You might also consider specifically referring to the microbe traits as 'microbe traits' and plant traits as 'plant traits' as I found this confusing in a few places.

“microbial traits”, is indeed a confusing term, was replaced with “rhizobiome traits” and specified in the introduction.

7. Line 505: remove 'however'.

Thanks. We have removed “However” in the revised text.

8. The final figure and analyses are exciting and very promising! Yet, you seem to shy away from emphasizing this point. I appreciate caution in drawing conclusions here but think that another sentence or two after line 431 would be appropriate.

To reinforce the main message of the manuscript, a final paragraph was added to the Results section as below.

“The example showcased here is a clear example of a three-way association between the abundance of a particular microbial group in the rhizosphere, a corresponding locus on the maize genome, and plant performance in the field. The datasets provided in this study contain several such associations and may serve as the basis for more targeted experiments to establish a direction of the causation chain from plant genotype to microbe abundance to plant performance, and to shed light on the genetic mechanisms that shape symbiotic relationships between the plant host and associated rhizosphere microbes.”

Reviewer #2 (Recommendations for the authors):This manuscript presents one of the largest 16S rRNA amplicon datasets I have ever seen. The study reports on a field experiment done with 230 maize genotypes under 2 nitrogen fertilization regimens. The abundance of each taxon within the root microbiota of over 3000 samples were used to perform two separate analyses: (i) a GWAS, using the abundance of each taxon as a quantitative trait and (ii) a phenotypic study, identifying microbial taxons correlated with different plant phenotypes (mainly canopy coverage).This is an impressive manuscript which will help pave the way for a better understanding of heritability in the plant microbiota, which is a major open question in our field. In massive screening studies such as this one, it is often challenging to deliver a concise take-home message or a mechanistic insight. Here, the actual functions of "MAPL" genes are not looked at in detail, besides the looking at their expression profiles across the plants (which is an interesting analysis in of itself). I therefore urge the authors to make their data, in the form of supplementary tables, as accessible as possible for follow up studies on the loci that they detected.

We thank the reviewer for this comment. The key data produced in this study is a list of associations between the abundance of microbial groups, corresponding plant genomic loci, and plant performance in the field. We provided five datasets as supplementary tables in the original submitted version, formatted for easy access, to enable and encourage more targeted experiments in the future. Additional metadata and associated phenotypic and genotypic information were annotated in the manuscript as clearly as possible. We also provided the datasets and scripts in the GitHub Repository (https://github.com/mandmeier/Maize_Rhizobiome_2022).

From what I can understand, and from some consultation with colleagues, the GWAS and related analysis were performed rigorously and with the appropriate normalizations and controls. However, I stress that I am not an expert in population genetics.Comments:One methodological aspect is unclear to me. How were the 150 "rhizobiome traits" defined? This seems like a taxonomic classification, but I am not clear how, or why, this was done? Also, the term "rhizobiome traits" is somewhat confusing, especially since these 150 vectors are referred to by different names throughout the manuscript. This is a key part of the analysis and should be explained clearly.

Thanks for the comments. A better explanation of the methodology used here was a point raised by all reviewers. We have added an additional section to the methods as well as an additional supplementary figure to explain the process in more detail (see response above for Reviewer #1 question 1).

A second and related comment, is why are the only "rhizobiome traits" that the authors consider taxon abundances? Other quantities can be extracted from this dataset: α diversity, β diversity (e.g the 1st and 2nd axes of the ordination). You could also try to extract a measure of absolute abundance from the ratio of bacterial to plastid/mitochondrial reads. These would provide a more holistic picture of how plant genotype influences the microbiome composition.

We agree that high-level rhizosphere microbial community metrics may be of interest to some readers. To this end, we performed additional GWAS analyses using four α diversity metrics (Observations, Shannon, Fisher, and Inverse Simpson) as well as the first 10 principal components. Key findings of this analysis are summarized in an additional supplementary figure:

A reference to this high-level analysis was added to the results.

“An initial analysis looking at high-level rhizobiome traits (Principal Components and α diversity metrics derived from the ASV table) shows the same pattern of divergent microbial communities between N treatments, and in particular under the -N treatment there is evidence for the association of plant genomic loci and microbiome composition (Supplementary Figure 9). To study changes in rhizobiome composition more accurately, the final 3,626 ASVs were clustered into n = 150 distinct microbial groups (“rhizobiome traits”), spanning 19 major classes of rhizosphere microbiota (Figure 1B, Supplementary Files 2 and 3) using a method previously described (Meier et al., 2021, Supplementary Methods).”

I am perceiving a bit of a disconnect between the GWAS and the phenotypic comparisons. Indeed, the authors show that heritability is correlated with the correlation with CC, but I would assume that much of the phenotypic differences observed in the first place are a result of the genotypic variability, but I do not see a unified model that accounts for the relationship between plant genotype, microbiome composition and plant phenotype. Am I missing something here?

Thanks for this insightful suggestion. A similar question was raised by Reviewer #1. We agree that the ultimate goal is to explain the relationship between plant genotype, microbiome composition, and plant phenotype in a unified model. However, to our knowledge, there is no such model readily available, and to develop such a three-variable model to connect genotype, microbiome, and plant phenotype altogether is challenging. Our group has attempted to build such a model for several years. For example, we leveraged a classical causal inference method, termed mediation analysis, to establish a causal chain from genotype to intermediate mediator to plant phenotype. We found the initial model can detect large effect mediators through extensive simulation. Using publicly available RNA-seq data as the mediators, we conducted empirical mediation analysis and identified mediator genes consistent with their known biological functions. As a next step, we will fit the microbiome data as the mediator. To get meaningful results, however, additional simulations and model fine-tuning need to be done. We believe the simulation and empirical results by fitting microbiome features as the intermediate mediators in a three-variable causal analysis is beyond the scope of this study.

For the current work, we opted to apply more established methods to investigate associations between microbe abundance and plant genetics, and between microbe abundance and plant performance.

The relative abundance and the prevalence of ASVs is essentially ignored, beyond the initial screening described in the appendix. It would be important to know for example if heritable taxa are also abundant ones?

We briefly reported the differential abundance of microbial groups in Supplementary Figure 2. It is an interesting consideration to test whether heritable taxa are also abundant ones. To do this, we plotted the heritability of each taxonomic group vs. the mean abundance and added an extra panel to Supplementary Figure 2. There is indeed a positive correlation between heritability and microbe abundance, suggesting that the low measured heritability of some low abundance microbes might result from the less precise quantification of abundance provided by smaller read counts for these taxa. Thank you for the suggestions!

Line 37: study, singular.

Corrected in the revised manuscript. Thanks.

Line 250: could the results of the PERMANOVA from the supplementary figure be highlighted also in the text?

We added the results of the PERMANOVA to the relevant section in the text.

Line 253: I do not understand how these 150 groups were classified. Also, why do you say that they are functionally distinct? All you have is their 16S sequence, there is no functional information (see comment above).

We understand this concern. To avoid confusion, we decided to refer to the groups identified here simply as “microbial groups”. The relevant sections in the manuscript were edited.

Figure 1b: As far as I could tell, the phylogenetic reconstruction in this tree does not match the consensus phylogeny for bacteria. This is more likely an error in their tree building algorithm than in the consensus tree and should be corrected.

This is a valid concern and more accurate phylogenetic clustering should be considered in future studies with emphasis on the evolution of plant-microbe associations. The phylogenetic tree used here serves only to illustrate the grouping of ASVs. Deviations from the consensus phylogeny were expected since only the 350bp ribosomal V4 region sequenced in this study was used to establish the relationship between groups. We added this caveat in the revised discussion.

Line 275: This could be measured (pagel's λ for example). This also raises the question why were all the ASVs binned to 150 groups?

The method used to bin ASVs into groups at low taxonomic ranks was explained in more detail (see reviewer #1, question 1 above). The resulting number of groups happened to be 150 and was not arbitrarily chosen.

Line 285: you didn't really frame this as a hypothesis.

Thanks. The wording was changed, as below to avoid confusion.

“A Bayesian-based (Genome-wide Complex Trait Bayesian analysis, or GCTB) method was used to test for signatures of selection for each rhizobiome trait (Materials and methods).”

Line 291: please explain the S parameter.

We clarified the text as below.

“Using the relationship between effects of non-zero SNPs and their minor allele frequencies (MAFs) as a proxy for the signature of selection (Zeng et al., 2018), the S value, a free parameter in the BayesS model, was estimated for both rhizobiome traits (Figure 2A) and plant traits (Figure 2B).”

Line 305: these 3 traits are not independent traits. More like 3 facets of the same trait.

Thanks for the comments. We agree that the leaf area, leaf fresh weight, and leaf dry weight are correlated traits. We added the pairwise correlation coefficients of these three traits in the revised manuscript.

Note that the three leaf-related traits are not independent. The pairwise correlation coefficients are 0.92, 0.91, and 0.94, for LA and FW, LA and DW, FW and DW, respectively.

Figure 3: why wasn't this analysis done for plant traits as well?

This analysis was done for both microbiome traits and plant traits. See Figure 2B for the results from the plant traits.

Line 505: "however" – I do not understand how this is contrary to the previous sentence.The terms "rhizosphere", "rhizobiome" and "root associated microbiome" seem to be used interchangeably here. Please sort this out.

We have removed “however” for clarity. “Root-associated microbiome” (microbial community) was changed to “rhizobiome” throughout the manuscript for consistency.

Line 779: was the DNA extraction kit substituted mid project? How did this affect the results?

The MagAttract PowerSoil kit and the KingFisher Flex purification system are two components of the same DNA isolation protocol. The text in the methods was edited for clarity as below:

“DNA was isolated from rhizosphere soil using the MagAttract PowerSoil DNA KF Kit (Qiagen, Hilden, Germany) and purified using the KingFisher Flex Purification System (Thermo Fisher, Waltham, MA, USA).”

Reviewer #3 (Recommendations for the authors):Comparing patterns of selection in N+ and N- environments can be done using methods that have been long established in evolutionary genetics: for example, calculating genotypic selection gradients in each of the treatments (Rausher 1992, https://doi.org/10.1111/j.1558-5646.1992.tb02070.x)

Thank you again for this insightful suggestion! According to the reviewer’s recommendations, we estimated selection gradients following the method developed by Lande and Arnold, 1983 and Rausher 1992 and recently implemented by Morrissey and Sakrejda.

Briefly, we estimated the fitness function relating fitness related traits, i.e., CC collected on August 22, to the abundance of the microbial groups with a generalized additive model (GAM). To reduce biases due to environmental covariances (Rausher 1992), we employed the BLUP values for both the microbial traits and the fitness related traits. Then, we obtained linear and quadratic selection gradients from the fitted GAM models using an R package developed by Morrissey and Sakrejda, 2013. We ran a total of 300 univariate models (150 microbial groups x 2 N treatments).

As a result, we found 58 microbial groups that showed significant selection gradients, summarized in a new supplementary Figure. To increase confidence in our results, we only reported selection data for 10 microbial groups that showed significant selection gradients AND significant selection coefficients in the GCTB analysis. Figure 2 was adjusted accordingly.

In theory, the patterns of heritability in rhizobiome features could be compared between treatments using a paired t-test.

We thank the reviewer for this idea. A paired Student’s t-test indeed shows a significant difference in heritability between the two N treatments. This finding was added to the results:

Revised text:

265: “Rhizobiome traits were comparatively more heritable under -N than +N conditions (paired Student’s t-test p = 0.021, Figure 1C).”